

# Understanding model biases in the diurnal cycle of evapotranspiration: a case study in Luxembourg

Maik Renner[1], Claire Brenner[2], Kaniska Mallick[3], Hans-Dieter Wizemann[4], Luigi Conte[1], Ivonne Trebs[3], Jianhui Wei[4], Volker Wulfmeyer[4], Karsten Schulz[2], Axel Kleidon[1]

[1]Max-Planck Institute for Biogeochemistry, Jena, 07745, Germany
[2]Institut für Wasserwirtschaft, Hydrologie und konstruktiven Wasserbau, Universität für Bodenkultur (BOKU), Wien, 1190, Austria
[3]Department Environmental Research and Innovation, Luxembourg Institute of Science and Technology (LIST), Belvaux, L-4422, Grand-duchy of Luxembourg
[4]Institut für Physik und Meteorologie, Universität Hohenheim, 70599 Stuttgart, Germany

*Correspondence to*: Maik Renner (mrenner@bgc-jena.mpg.de)

**Abstract.** The diurnal forcing of solar radiation is the largest signal within the Earth system and dominates the diurnal cycle of the turbulent heat fluxes and evapotranspiration ($\lambda E$) over land. Incoming solar radiation ($R_{sd}$) also shapes temperature, vapor pressure deficit and wind speed known as important controls on λE. Current process-based $\lambda E$ schemes used in remote sensing and land-surface modeling differ in how these controls on $\lambda E$ are represented and which input variables are required. Here, we analyze how well different surface energy balance schemes are able to reproduce the diurnal cycle and how the diurnal signals of observed input variables actually influence the resulting diurnal pattern of $\lambda E$. As additional constraint for model evaluation we estimate a linear and a non-linear phase shift component of a surface variable (e.g. $\lambda E$) to incoming solar radiation. We illustrate our analysis with observations from an eddy covariance station at a temperate grassland site in Luxembourg with a focus on clear sky conditions. During the field campaign in 2015 a summer drought led to a dry-down of soil moisture which allows for studying the effect of wet and dry conditions on the diurnal cycle.

We found a remarkable, almost linear relationship of $\lambda E$ with $R_{sd}$, which exhibits a significant positive phase lag during wet periods. This phase lag in $\lambda E$ was compensated by a preceding phase lag of the sensible heat flux. Vapor pressure deficit ($D_a$, often used as input for Penman-Monteith based approaches) exhibits a strong phase lag, which is driven by air temperature reflecting large diurnal heat storage changes in the lower atmosphere. This large phase lag in $D_a$, which is not seen in $\lambda E$, explains why actual and potential evapotranspiration approaches can show systematic deviations from observations at the sub-daily time scale and highlight the need for a time-dependent non-linear compensation through the conductance parameterization. The surface to air temperature gradient used as input in energy balance residual approaches corresponds rather well with its linear response and phase lag to the observed sensible heat flux under both, wet and dry conditions. This simplifies the conductance parameterization and explains the better correlation of these models at the sub-daily time scale.

We conclude that the analysis of phase lags at the sub-daily time-scale provides valuable information on the drivers of the surface-atmosphere exchange, which can be used to evaluate and improve the representation of land-atmosphere coupling in land-surface schemes.



## 1 Introduction

Evapotranspiration and the corresponding latent heat flux ($\lambda E$) couple the surface water and energy budgets and are of high relevance for water resources assessment. $\lambda E$ is generally limited by four physical factors, (i) the availability of energy mostly supplied by solar radiation, (ii) the availability of and the access to water, (iii) the plant physiology, and (iv) the

atmospheric transport of moisture away from the surface (Brutsaert 1982). These different limitations have led to different approaches on how to model $\lambda E$.

Key approaches either focus on the surface energy balance where the surface-air temperature gradient dominates the flux; or approaches which focus on the moisture transfer limitation where vapor pressure gradients dominate the flux. It is critical to

recognize that these two limitations are not independent of each other but rather are shaped by land-atmosphere heat and water exchange and thus covary with each other. The diurnal variation of incoming solar radiation ($R_{sd}$) causes a strong diurnal imbalance in surface heating leading to the pronounced diurnal cycles of surface states and fluxes (Oke 1987, Kleidon and Renner, 2017). This heat exchange of the surface with the lower atmosphere thus influences the near-surface air temperature ($T_a$), skin temperature ($T_s$), vapor pressure ($e_a$), soil or canopy saturation water pressure ($e_s$), vapor pressure

deficit ($D_a$), and wind speed ($u$), that are being regarded as important controls on $\lambda E$ (e.g. Penman 1948). These interactions are particularly dominant at the diurnal time scale (e.g. De Bruin and Holtslag 1982) and depend on meteorological as well as on surface conditions (Jarvis and McNaughton 1986). Ignoring the interdependence of the surface variables may lead to biases in model parameterizations and compensating errors when evaluating the model performance only with respect to a single variable (Matheny et al., 2014, Best et al., 2015, Santanello et al., 2018).

To overcome these problems several authors suggested to better evaluate land-atmosphere (L-A) interactions proposing different multivariate metrics (e.g., Betts 1992, Santanello et al. 2009, Matheny et al., 2014, Zhou et al., 2016b, Santanello et al., 2017). These metrics explore internal relationships between state variables to better characterize key processes and to guide a more systematic exploration and understanding of model deficiencies. There is a strong need to investigate and to

derive metrics based on comprehensive observation that characterize the whole land surface-atmosphere system (Wulfmeyer et al. 2018).

Here, we propose an alternative metric, which focuses on the diurnal variation of surface states and fluxes with respect to incoming solar radiation ($R_{sd}$). It is well established that $R_{sd}$, which shapes the diurnal cycle of net radiation ($R_n$) over land, is

strongly correlated with $\lambda E$. Examples are the successful application of equilibrium evapotranspiration (Schmidt 1915, Priestley-Taylor 1972, Miralles et al., 2011, Renner et al., 2016), the semi-empirical Makkink equation (Makkink 1957, De Bruin and Lablans 1997, De Bruin et al., 2015) and using a fixed fraction of available energy for estimating potential evapotranspiration (Milly and Dunne 2016, Maes et al., 2018). Observations of a near linear relationship of $\lambda E$ to $R_{sd}$



(Jackson et al., 1983) and the need for spatial mapping of $\lambda E$ stimulated remote sensing based applications (Crago, 1996). Although observations justify this simple behavior, the role of other controls on $\lambda E$ remains unclear (De Bruin and Holtslag 1982, Beljaars and Bosveld 1997, Best et al., 2015; Haughton et al., 2016; Zhou and Wang 2016a; Milly and Dunne 2016). This is particularly important for remote sensing based $\lambda E$ models since they use observational states which have an imprint 5 of land-atmosphere interactions.

The relevance of other atmospheric controls on $\lambda E$ should result in a deviation from linearity between $\lambda E$ and $R_{sd}$. There could be a consistent lag between the diurnal patterns of $\lambda E$ versus $R_{sd}$ which would reveal a loop when $\lambda E$ is plotted against solar radiation. There is observational evidence of such consistent loops for heat fluxes plotted against net radiation 10 (Camuffo and Bernadi 1982; Mallick et al., 2015), with many studies showing hysteretic loops of the soil heat flux against net radiation (Fuchs and Hadas, 1972; Santanello and Friedl, 2003; Sun et al., 2013). Camuffo and Bernardi (1982) showed that the magnitude and direction of such hysteretic loops can be estimated by a multi-linear regression of the variable of interest against the forcing variables and its first order time-derivative. This simple model allows to estimate storage effects on diurnal (Sun et al. 2013) to seasonal time scales (Duan and Bastiaansen 2017).

Here we use $R_{sd}$ as independent forcing of land-atmosphere exchange to quantify the response of surface heat fluxes and states to $R_{sd}$ as well as the magnitude and direction of phase lags. Thereby we obtain a metric to measure the influence of the various drivers of $\lambda E$. Process-based models represent our knowledge of the various drivers of $\lambda E$ and therefore should be able to capture the magnitude of hysteretic loops under different conditions (Matheny et al., 2014). Hence, we can use the 20 hysteretic behavior as a pattern-based approach and as an additional criterion to evaluate the performance of different models. We focus on two different approaches to estimate $\lambda E$. The first approach is based on the energy limitation of $\lambda E$, using the equilibrium evaporation concept (Schmidt 1915) as formulated by Priestley and Taylor (1972) for potential evaporation. For actual evaporation we focus on one source and two source energy balance schemes (OSEB and TSEB, respectively) which derive $\lambda E$ as residual term of the surface energy balance and parameterize the sensible heat flux by a 25 temperature gradient - resistance description (Kustas and Norman 1996) (referred to as 'temperature gradient scheme'). The second approach is based on the Penman-Monteith (PM hereafter) approach (Monteith 1965), which adds water vapor pressure deficit as a driving gradient (referred to as 'vapor gradient scheme'). We use the widely used FAO Penman-Monteith formulation (Allen et al., 1998) for potential or reference evapotranspiration. For actual evapotranspiration we use a modified PM approach which was formulated by Mallick et al. (2014, 2015, 2016, 2018); (see also Bhattarai et al., 2018) 30 and is termed as a surface temperature initiated closure (STIC). STIC is based on finding the analytical solution of the surface and aerodynamic conductances in the PM equation while simultaneously constraining the surface and aerodynamics conductances through both surface temperature and vapor pressure deficit.



Several inter-comparison studies evaluated the performance of these schemes using observations from different landscapes. OSEB and TSEB which are often used in remote sensing retrievals of $\lambda E$ have been found to perform comparably well in reproducing tower-based energy flux observations (Timmermans et al. 2007; Choi et al. 2009; French et al., 2015). Yang et al. (2015) compared temperature gradient approaches (including TSEB) with the Penman-Monteith approach (based on

vapor pressure gradient only) employed by the MODIS evapotranspiration product (MOD16, Mu et al., 2011) and found strongly reduced capability of MOD16 to estimate spatial variability of evapotranspiration. They concluded that the moisture availability information obtained from relative humidity and vapor pressure deficit is not able to capture the surface water limitations as reflected in surface temperature.

In this study, we focus on the ability of these different evapotranspiration models to reproduce the diurnal cycle of $\lambda E$ under wet and dry conditions. In particular, we assess if significant non-linear relationships in form of hysteretic loops exist, if these change under different wetness conditions and if temperature-gradient and vapor-gradient approaches such as PM are able to reproduce this behavior. Further, we evaluate which input variables of the evapotranspiration schemes show a hysteretic pattern and how these patterns influence the flux estimation. To address these questions, we analyze observations

and models with respect to internal functional relationships (pattern-based) and use solar radiation as independent driver of land-atmosphere exchange. We focus on wet vs. dry conditions since this is another critical deficiency identified in previous analyses (e.g. Wilson et al., 2003, Matheny et al., 2014, Zhou et al., 2016b). To ensure similar radiative forcing and avoid variability due to cloud cover we focus the evaluation on clear-sky days. We illustrate our approach on a grassland site in a temperate semi-oceanic climate using surface energy balance observations.

The analysis will shed light on the capabilities of process-based evapotranspiration schemes to capture the dynamics of diurnal land-atmosphere exchange. We show that the phase lag of surface states and fluxes reveals important imprints of heat storage processes and how this guides the evaluation of the different approaches for modeling $\lambda E$. This is important for applications in remote sensing with respect to the choice of observational input variables. In doing so, we provide a further,

pattern-based metric to assess land-atmosphere interactions, and, thus guide process-based improvements and calibration of land-surface schemes.

## 2 Methods and Data

### 2.1 Diurnal patterns and hysteresis loop quantification

We first illustrate the pattern-based evaluation of the diurnal cycle using two hypothetical variables $Y_1$ and $Y_2$, as shown in

Figure 1. If a variable ($Y_1$) is in phase with $R_{sd}$, it shows a linear behavior when plotted against $R_{sd}$ (Fig. 1 b). However, if a variable ($Y_2$) has a time lag with respect to $R_{sd}$, showing a significant difference between morning and afternoon values, it results in a hysteretic loop. The area inside the loop indicates the magnitude of the phase difference, while the direction of





the loop, marked by an arrow at the morning rising limb in Fig. 1b, indicates if a variable is preceding or lagging $R_{sd}$ in time. If a variable shows consistently larger values during the afternoon as compared to the morning, this will appear as a counter-clockwise (CCW) hysteretic loop indicating a positive phase lag with respect to $R_{sd}$. A negative phase lag appears as a clockwise (CW) loop.

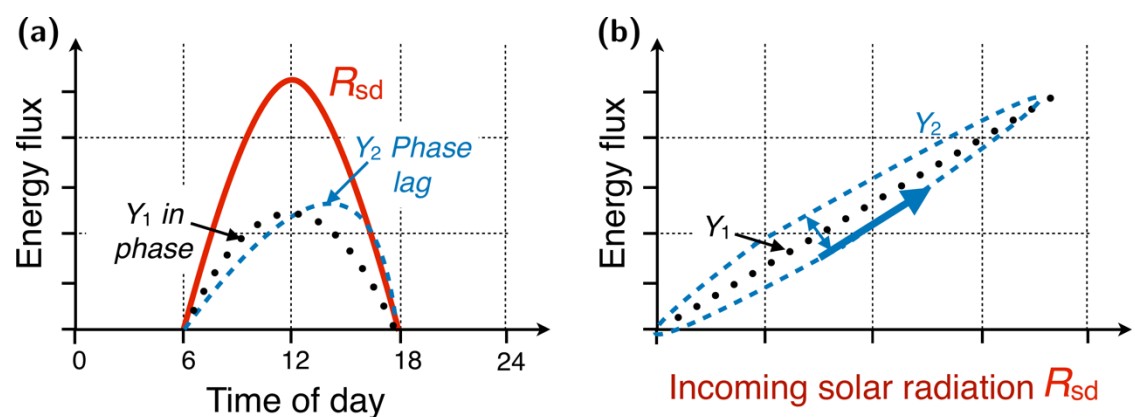

**Figure 1: Illustration of a pattern-based evaluation of the diurnal cycle. Panel a) shows the diurnal cycle of $R_{sd}$ under clear-sky conditions and the diurnal cycle of two variables $Y_1$ and $Y_2$, one in phase with $R_{sd}$ and another lagging $R_{sd}$. Panel b) illustrates the relationship of these variables when plotted against $R_{sd}$. The bold arrow indicates the direction of the loop and the area inside the**
**hysteresis describes the magnitude of the phase shift.**

To obtain a quantitative measure of the hysteretic pattern, we use the Camuffo-Bernardi equation (Camuffo and Bernardi 1982), which relates the time series of the response variable $Y(t)$ to the forcing variable $R_{sd}(t)$ and its first order time derivative $dR_{sd}(t)/dt$:

$Y(t) = a + b\, R_{sd}(t) + c\, (dR_{sd}(t)/dt) + \varepsilon(t).$                                                                      (1)

Using multi-linear regression, we obtain the coefficients $a$, $b$ and $c$ assuming a normal distribution of the residuals $\varepsilon(t)$. If $Y$ is linear with $R_{sd}$, the parameter $c$ should be zero. However, if a consistent pattern such as a hysteretic loop exists, then parameter $c$ should be significantly different from zero. Hence, by using regression analysis we can determine if a significant

hysteretic relationship between two variables exists, and if the inclusion of such a non-linear term (with $c \neq 0$) would improve the model fit.

Both, the parameters $b$ and $c$ contain information on the phase lag between the two variables. Assuming that $R_{sd}$ is a harmonic function with an angular frequency $\omega$, the phase difference can be estimated by:

$\varphi = \tan^{-1} (c\ \omega\ /\ b)$                                                                                              (2)





To derive the first order time derivative of solar radiation, we use a simple difference between time steps. Since the data we use is available in 30 min time steps (see below), we have 48 time steps per day, thus $\omega = 48/(2\pi)$. To obtain a phase lag between $Y$ and $R_{sd}$ as a time lag $t_\varphi$ [min] we use:

$$t_\varphi = \tan^{-1} (48/(2\pi) \; c \, / \, b) \, (60 \times 24/(2\,\pi)). \tag{3}$$

Note that the phase lag estimate $t_\varphi$ is somewhat similar to the relative diurnal centroid metric proposed by Wilson et al., (2003) for the analysis of the timing of heat and mass fluxes. The diurnal centroid identifies the timing of the peak of a variable with respect to local time. Since the peak of $R_{sd}$ is at noon local time both metrics are qualitatively comparable.

## 2.2 Field site and observations

The study area is a grassland site in Petit-Nobressart, Luxembourg, situated on a gentle east facing slope. The grassland is used as a hay meadow and had short vegetation of about 10-15 cm as the grass was mowed before the start of the experiment. An Eddy Covariance (EC) station (with the setup described in Wizemann et al. 2015) was installed at the grassland close to the village of Petit Nobressart (Fig. 2, exact coordinates: N 49° 46.77' E 05° 48.22'). The EC station was operated from 11 June until 23 July 2015. The three-dimensional wind and temperature fluctuations were measured at 2.41 m above ground by a sonic anemometer (CSAT3, Campbell Scientific Inc., Logan, USA) facing to the mean wind direction of 290°. A fast response open-path $CO_2$/$H_2O$ infrared gas analyzer (IRGA LI-7500, LI-COR, USA) installed in a lateral distance of 0.2 m to the sonic path was used to measure $CO_2$ and $H_2O$ fluctuations. The high-frequency signals were recorded at 10 Hz by a CR3000 data logger and the TK3 software was used to compute turbulent fluxes of sensible heat ($H$), latent heat ($\lambda E$) and $CO_2$ (Mauder and Foken, 2015).

Downwelling and upwelling shortwave and longwave radiation were obtained by a four-component net-radiation sensor (NR01, Hukseflux, Netherlands). The meteorological variables (air temperature, humidity and precipitation) were monitored with a time resolution of 30 minutes. Sensors measuring soil heat flux (two in 8 cm depth, HFP01, Hukseflux, Netherlands), soil temperature (2, 5, 15, 30 cm, model 107, Campbell Scientific Inc., UK), water content (2.5, 15, 30 cm, CS616, Campbell Scientific Inc., UK) and matric potential (5, 15, 30 cm, model 253, Campbell Scientific Inc., UK) were installed between the turbulence and radiation measurement devices.

Unfortunately, the two upper temperature probes and matric potential sensors showed data gaps and erroneous values from 30 June until excavation on 23 July, 2015. Thus, the ground heat flux was calculated by the heat flux plate method with correction for heat storage (Massman, 1992) only for the period from 11 June to 30 June, 2015. To still obtain soil heat fluxes for the entire measuring period, additionally harmonic wave analysis (Duchon and Hale, 2012)) of the heat flux plate data was applied. Both methods agreed well for the period before 30 June, so that the latter method should provide reliable ground heat flux values for the entire period until 23 July. Table 1 lists the variables obtained from EC station and used





in this work. For more details on instrumentation and EC data processing see Ingwersen et al. (2011) and Wizemann et al. (2015).

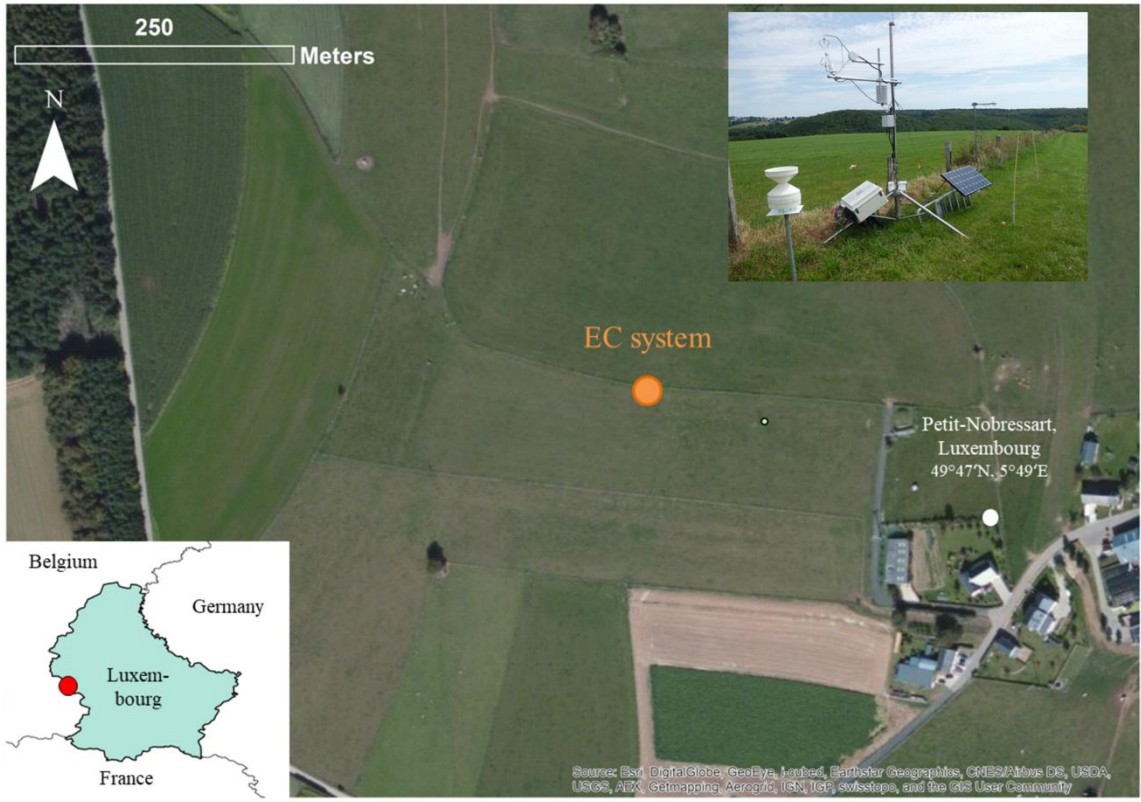

**Figure 2: Location of the EC site at Petit-Nobressart, Luxembourg. Top right inset shows the mast with sonic anemometer and infrared gas sensor (center), radiation sensor (right) and rain sensor (left). The soil sensors are located on the right of the solar panel. Photo: Elisabeth Thiem. Background: ESRI ® World Imagery.**

### 2.2.1 Derived meteorological variables

We derived the saturated water vapor pressure $e_s$ (hPa) by the empirical Magnus equation (Magnus 1844) as a function of air temperature $T$ (°C):

$$e_s(T) = 6.1078 \text{ hPa e}^{(17.08085\, T\, /\, (234.175\, +\, T))}$$



Then, the water vapor pressure of the air $e_a$ (hPa) was obtained by using air temperature $T_a$ and relative humidity ($r_H$):

$$e_a = e_s(T_a)\, r_H/100.$$

To assess the moisture conditions of each date of the site we used the evaporative fraction $f_E$:

$$f_E = \lambda E\,/\,(H + \lambda E).$$

Since daily averages can be influenced by single large values of the turbulent fluxes and contain missing values, we
estimated a daily $f_E$ based on the 30 min values of each day using the following linear regression:

$$\lambda E = f_E\,(H + \lambda E) + \alpha + \varepsilon_R$$

where $f_E$ is the slope of the linear regression, $\alpha$ its intercept and $\varepsilon_R$ the residuals. Since we use the fluxes of $H$ and $\lambda E$ without
energy balance closure correction we obtain the upper range of $f_E$.

Since the sonic anemometer measures friction velocity ($u*$) and the absolute value of wind speed $u = sqrt(U^2+V^2)$, we
estimate the aerodynamic conductance for momentum ($u*^2/u$) and the aerodynamic conductance ($g_{ah}$) for heat including the
excess resistance to heat transfer using an empirical formula by Thom (1972):

$$g_{\text{ah,Thom}} = \left(\frac{u}{u^{*2}} + \frac{6.2}{u^{*\frac{2}{3}}}\right)^{-1} \tag{4}$$

We chose to use this formula for its simplicity and similar performance than more recent, complex parameterizations
(Knauer et al., 2018). Also note that effects of atmospheric stability are accounted for in the first term of Eq. (4).

**2.2.2 Energy balance closure gap correction**

Most EC measurements show that the sum of the observed turbulent heat fluxes are smaller than the available energy and
thus do not close the energy balance leaving an energy balance closure gap ($Q_{gap}$) (Foken et al., 2008)

$$Q_{\text{gap}} = R_n - (G + H + \lambda E)$$

For our site we observed on average $(H + \lambda E)\,/\,(R_n - G) = 0.81$ with an average gap of 67 W m$^{-2}$ over the whole duration of
the field campaign. These values are in the typical range what is commonly found for grassland sites (Stoy et al., 2013).



To correct the turbulent fluxes for the energy balance closure gap, we use a correction based on the Bowen ratio ($B_R$) (Twine et al., 2000), which is directly related to $f_E = 1/(B_R+1)$ to obtain corrected fluxes:

$\lambda E_{BRC} = \lambda E + Q_{gap} * f_E$

and

$H_{BRC} = H + Q_{gap} * (1\text{-}f_E)$

We use these corrected fluxes in the further analysis.

### 2.2.3 Clear-sky day classification

In order to achieve comparable conditions with respect to incoming solar radiation, we identified clear-sky conditions. A clear-sky day was defined by its daily sum of incoming solar radiation being larger than 85% of the potential surface

radiation ($R_{sd,pot}$), which is a function of latitude and day of year (using R package REddyProc, function fCalcPotRadiation):

$R_{sd}/R_{sd,pot} > 0.85 \ \Sigma(R_{sd}(t)) / \ ( \ f_{diff} \ \Sigma(R_{sd,pot} \ (t))),$

where t corresponds to each time step of measurement and $f_{diff} = 0.78$ being a constant factor taking account for atmospheric

absorption of solar radiation.

### 2.3 One and Two Source Energy Balance models

Thermal remote sensing based models estimate evapotranspiration by solving the surface energy balance and rely on land surface temperature ($T_s$) information as a key boundary condition (Kustas and Norman 1996). A bulk layer formulation of the soil-plus-canopy sensible heat flux is employed and $\lambda E$ is derived by enforcing the surface energy balance. Hence $\lambda E$ is

written as:

$\lambda E = R_n - G - H = R_n - G - \rho \, c_p \, (T_s - T_a) \, g_{ah} \,,$    (5)

where $\rho$ is density of air, $c_p$ is the specific heat of air at constant pressure, and $g_{ah}$ is the effective aerodynamic conductance

of heat that characterizes the transport of sensible heat between the surface and the atmosphere. We obtained $T_s$ from the observed longwave emission of the surface $T_s = (R_{lu}/(\sigma \, \varepsilon_s))^{1/4}$ with $\sigma = 5.67 \times 10^{-8}$ W K$^{-4}$ the Stefan Boltzmann constant and a surface emissivity $\varepsilon_s = 0.98$, which is typical for a grassland and agrees with Brenner et al., (2017).





We use two different approaches which are generally classified as one- and two-source models with regard to the implemented treatment of the energy exchange with the surface. While one-source energy balance models (OSEB) treat the surface as a uniform layer, two-source energy balance models (TSEB) partition temperatures, radiative and energy fluxes into a soil and vegetation component. The one-source approach (OSEB) parameterizes the aerodynamic conductance $g_{ah}$ as

follows (e.g. Kalma et al., 2008, Tang et al., 2013):

$$g_{ah,\mathrm{OSEB}} = \frac{k^2 u}{[\ln((z_u-d)/z_{0m})-\Psi_m]\,[\ln((z_t-d)/z_{0m})+\ln(z_{0m}/z_{0h})-\Psi_h]} \tag{6}$$

where $z_u$ and $z_t$ are the measurement heights of wind and air temperature, respectively, $z_{0m}$ and $z_{0h}$ are roughness lengths for

momentum and heat, respectively, $k$ is the von Kármán constant, $d$ is the displacement height, $u$ is the wind speed and $\Psi_m$ and $\Psi_h$ are the the integrated Monin-Obukhov (MO) similarity functions which correct for atmospheric stability conditions (Brutsaert 2005, Jiménez et al., 2012). For the investigated grassland site, $d$ and $z_{0m}$ were calculated as fractions of the vegetation height, $h_c$, with $d = 0.65\,h_c$ and $z_{0m} = 0.125\,h_c$. The roughness length for heat $z_{0h}$ was set using the dimensionless parameter $kB^{-1} = \ln(z_{0m}/z_{0h})$, which was set to 2.3 in accordance with Bastiaanssen et al., 1998. Note that this

parameterization of aerodynamic conductance does not explicitly distinguish between bare soil and canopy boundary layer conductance, as it is done in two-source approaches.

In addition to OSEB we applied the Two-Source Energy Balance (TSEB) model developed by Norman et al., (1995), Kustas and Norman (1999). For both, the soil and canopy components a separate energy balance (with different component

temperatures) and bulk resistance scheme with different aerodynamic conductance are formulated. Then the energy balance equations are solved iteratively. It starts by assuming that a fraction of the canopy (described by vegetation greenness fraction $f_g$) transpires at a potential rate as described by the Priestley-Taylor equation (Priestley and Taylor 1972):

$$\lambda E_{PT} = \alpha_{PT}\frac{s}{s+\gamma}R_n \tag{7}$$

where $\alpha_{PT}$ is the Priestley-Taylor coefficient (1.26), $s$ is the slope of the saturation water vapor pressure curve and $\gamma$ is the psychrometric constant. However, the canopy latent heat flux $\lambda E_c = f_g\,\lambda E_{PT}$ might be too large and the soil component would become negative (condensation at the soil surface) which is unlikely during daytime conditions. To avoid condensation at the soil surface, the $\alpha_{PT}$ coefficient is reduced incrementally until the soil latent heat flux becomes zero or positive. Once this

condition is met, all other energy balance components are updated accordingly to satisfy the energy balance equation. For this study we used a constant vegetation fraction of $f_c = 0.9$ and a greenness fraction $f_g$ which was derived from close-up pictures taken at the beginning and the end of the field campaign and linearly interpolated in-between.



## 2.4 Penman-Monteith approach

In the Penman-Monteith approach (Monteith 1965) the inclusion of physiological conductance ($g_s$) imposes a critical control on $\lambda E$:

$$\lambda E = \frac{s\,(R_n - G) + \rho\,c_p g_{av}(e_s(T_a) - e_a)}{s + \gamma(1 + \frac{g_{av}}{g_s})}$$
(8)

In equation (8), the transfer of moisture is linked to a supply-demand reaction where the net available energy ($R_n - G$) is the supplies the energy for evaporation and the vapor pressure deficit of the air, $D_a$ [$= e_s(T_a) - e_a$] is the demand for evaporation from the atmosphere. In the PM approach, the two conductances, the aerodynamic conductance $g_{av}$ and the surface

conductance $g_s$ to water vapor are unknown. A widely used approach to obtain a reference evapotranspiration estimate from meteorological data is the FAO Penman-Monteith reference evapotranspiration (Allen et al., 1998). It defines the two conductances for a well-watered grass surface with a standard height of $h = 0.12$ m. The aerodynamic conductance is obtained by a bulk approach (eq. 6) with wind speed $u$ measured at 2m above the surface, $d = \frac{2}{3}\,h$, $z_{0m} = 0.123h$, $z_{0h} = 0.1z_{0m}$ yielding $g_{av} = u/208$ (Box 4 in Allen et al., 1998). Surface conductance is fixed at a constant $g_s = 1/70$ *m/s*. While the FAO

estimate is typically intended for estimates of the reference evaporation for well-watered grass on a daily basis, we use it here as a reference for comparison on a subdaily scale. In order to understand the effect of the aerodynamic conductance parameterizations we add another reference evapotranspiration estimate in which the aerodynamic conductance is given by eq. (4) using observations of friction velocity and wind speed, but keeping $g_s$ fixed.

### 2.4.1 Penman-Monteith based Surface Temperature Initiated Closure (STIC) (version STIC1.2)

In order to estimate an actual evapotranspiration rate from meteorological data we employ a method (STIC1.2 hereafter referred to as STIC), which is based on the PM equation, but which in addition integrates surface temperature information. The STIC methodology is based on finding analytical solutions for the two unknown conductances to directly estimate $\lambda E$ (Mallick et al., 2016, 2018). STIC is a one-dimensional physically-based SEB model that treats the vegetation-substrate complex as a single unit (Mallick et al., 2016; Bhattarai et al., 2018). The fundamental assumption in STIC is the first order

dependency of $g_a$ and $g_s$ on soil moisture through $T_s$ and on environmental variables through $T_a$, $D_a$, and net radiation. Thereby, surface temperature is assumed to provide information on water-limitation which is linked to the advection-aridity hypothesis (Brutsaert and Stricker 1979). In STIC, no wind speed is required as input data, as opposed to the temperature gradient approaches, but vapor pressure of the air and its saturation value become critical input variables, see Table 2 for an overview. A detailed description of STIC version 1.2 is available in Mallick et al. (2016, 2018) and Bhattarai et al. (2018).





# 3 Results

## 3.1 Daily clear-sky and moisture classification

The field campaign was conducted during an exceptionally warm and dry period characterized by clear sky conditions with remarkably high air temperatures with daily maxima above 30°C and little precipitation. Compared to the climatic normal

(1981-2010) the precipitation deficit in this region was -44% in June and -41% in July, respectively (source: meteorological station Arsorf, Administration des services techniques de l'agriculture (ASTA)). The air temperature anomaly was higher in July (1.9°C) than in June (0.7°C) (source: meteorological station Clemency, ASTA). The soil water content decreased and parts of the site, especially in the upper part, showed clear signs of vegetation water stress (see Brenner et al., (2017) for an analysis of the spatial heterogeneity of water limitation). However, the dry period was interrupted by a few but strong

rainfall events, which significantly changed soil moisture and thus $f_E$ with time (Fig. 3a). Based on the observed $f_E$ we classified dry days with $f_E < 0.5$ and wet days with $f_E > 0.6$. This separation of dry and wet days is also reflected in the top soil moisture conditions (measured at 5 cm depth) as shown in Fig 3b.

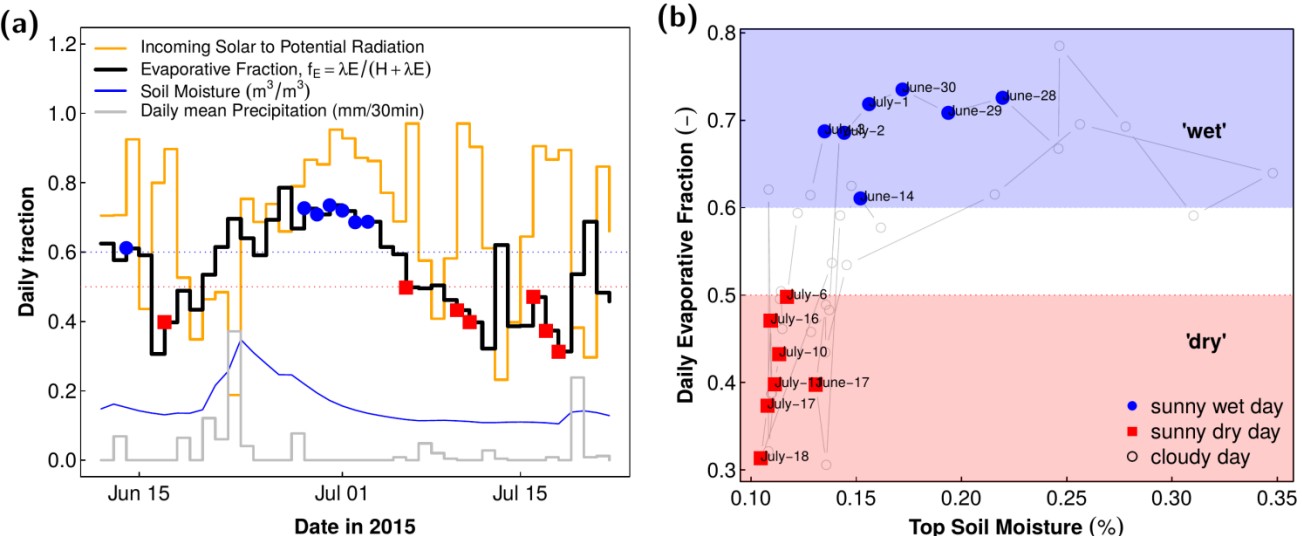

**Figure 3: Daily observations of soil moisture, evaporative fraction, ratio of observed to potential solar radiation and mean precipitation. Panel a) shows the daily time series and panel b) the relationship of $f_E$ to soil moisture used to classify "wet" and "dry" days depending on $f_E > 0.6$ or $f_E < 0.5$, respectively. Sunny days are defined using a threshold of 85% of $R_{sd}$ to potential radiation and are marked with solid symbols, with blue circles referring to wet and red squares to dry days. Top soil moisture measured at 5 cm below surface is shown.**

Based on the classification into wet and dry days under clear-sky conditions we computed composites of the diurnal cycle for each hour. By using only sunny days we aim to achieve similar conditions with respect to downwelling shortwave radiation ($R_{sd}$). Figure 4a confirms that $R_{sd}$ and net radiation ($R_n$) had very similar diurnal cycles and magnitudes for the wet





and dry days. However, the downwelling longwave radiation $R_{ld}$ and the soil heat flux were somewhat higher under wet conditions (Fig. 4a). The higher $R_{ld}$ is related to a higher air temperatures and air vapor pressures observed under wet conditions (Fig. 4b), which may explain the greater value of $R_{ld}$ by affecting the atmospheric emissivity for longwave radiative exchange. This has also an impact on the minimum temperatures both for air and skin temperature, which are

higher under wet conditions and lower under dry conditions (Fig. 4b). Hence, although we achieve fairly similar conditions for shortwave radiation under wet and dry conditions, we observed a small but significant difference in the longwave radiative exchange.

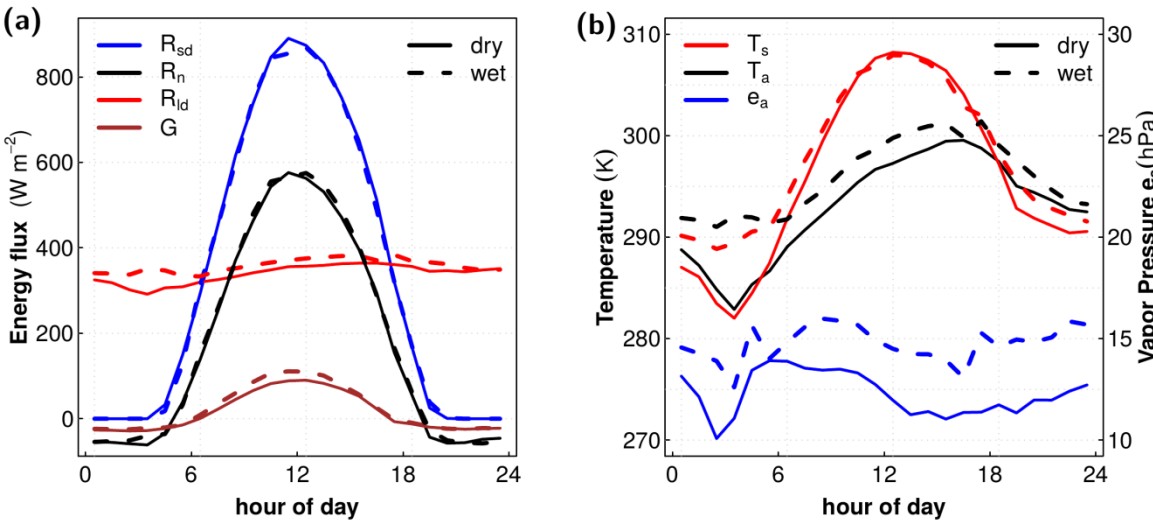

**Figure 4: Observations of average diurnal cycles of energy fluxes (panel a, with $R_{sd}$: shortwave downwelling flux, $R_{ld}$: longwave downwelling flux; $R_n$: net radiation; $G$: ground heat flux); surface and air temperatures, $T_s$ and $T_a$, and air vapor pressure, $e_a$, (panel b) comparing wet and dry days.**

### 3.2 Diurnal cycle of evapotranspiration under wet and dry conditions

Next, we evaluate how the different evapotranspiration schemes are able to reproduce the fluxes during wet and dry

conditions under similar $R_{sd}$ forcing. Figure 5 shows the average diurnal cycle of observations and models for $\lambda E$. The observations showed a significant difference in $\lambda E$ between dry and wet conditions, with the maximum value of $\lambda E$ of about 200 W m$^{-2}$ for dry and 350 W m$^{-2}$ under wet conditions, which amounts to a mean difference of 100 W m$^{-2}$ for daylight conditions (Table 3). As reference, we also included two common formulations of potential evapotranspiration, the Priestley-Taylor potential evapotranspiration (PT) and the FAO Penman-Monteith reference evapotranspiration (FAO-PM). Both do

not account for water limitation and show a marginal difference of 10 W m$^{-2}$ between wet and dry conditions. While FAO-PM yielded lower mean conditions than PT, it showed lower correlation and RMSE as compared to PT (Table 3). We find that all models for actual $\lambda E$ (rather than PT or FAO-PM) showed differences in $\lambda E$ between wet and dry conditions. Both OSEB and TSEB showed a tendency to overestimate $\lambda E$ under dry conditions but captured the high $\lambda E$ values under wet





conditions. In contrast, STIC captured the low $\lambda E$ magnitude under dry conditions ($f_E<0.5$) but underestimated $\lambda E$ under wet conditions (for $f_E>0.6$).

Table 3 shows the statistical metrics of the model performances with respect to the Bowen ratio corrected $\lambda E$. In general, both OSEB and TSEB produced mean $\lambda E$ values within the range of 96 - 98% (255 W m$^{-2}$ and 259 W m$^{-2}$) of the observed $\lambda E$ (264 W m$^{-2}$) in wet conditions, while mean $\lambda E$ from STIC was within 83% (218 W m$^{-2}$) of observed $\lambda E$ for the same conditions. However, for the dry conditions, simulated $\lambda E$ from STIC (180 W m$^{-2}$) was 91% of the observed mean $\lambda E$ (164 W m$^{-2}$), while the simulated mean $\lambda E$ from OSEB and TSEB was 77 - 78% of the observed mean $\lambda E$. Overall, the three models captured 86% (OSEB), 88% (TSEB), and 95% (STIC) of the observed mean $\lambda E$. Results show that under wet conditions, RMSE of the OSEB /TSEB models is well within the range of the errors when compared with the uncorrected $\lambda E$, whereas STIC showed relatively higher RMSE. However, under dry conditions the RMSE of OSEB /TSEB models was found to be larger than for STIC. For the entire observation period the three models produced comparable RMSE (41 - 46 W m$^{-2}$) but with different correlation. STIC produced relatively low correlation (r$^2$ = 0.72) as compared to the other two models (r$^2$ = 0.84 – 0.85). Thereby, we find that the correlation of the schemes is distinctly larger under wet conditions as compared to dry conditions. The correlation under wet conditions of OSEB and TSEB are in the range of the correlation of the uncorrected $\lambda E$ (r$^2$ = 0.91), whereas STIC and FAO-PM showed lower correlation. Under dry conditions the correlation was significantly lower than the correlation of the uncorrected $\lambda E$ (r$^2$ = 0.93). While OSEB/TSEB explained 62% of the observed uncorrected $\lambda E$ variability in dry conditions (STIC explained 44%), both models produced higher RMSE (57 - 58 W m$^{-2}$) as compared to STIC (45 W m$^{-2}$) under these conditions.





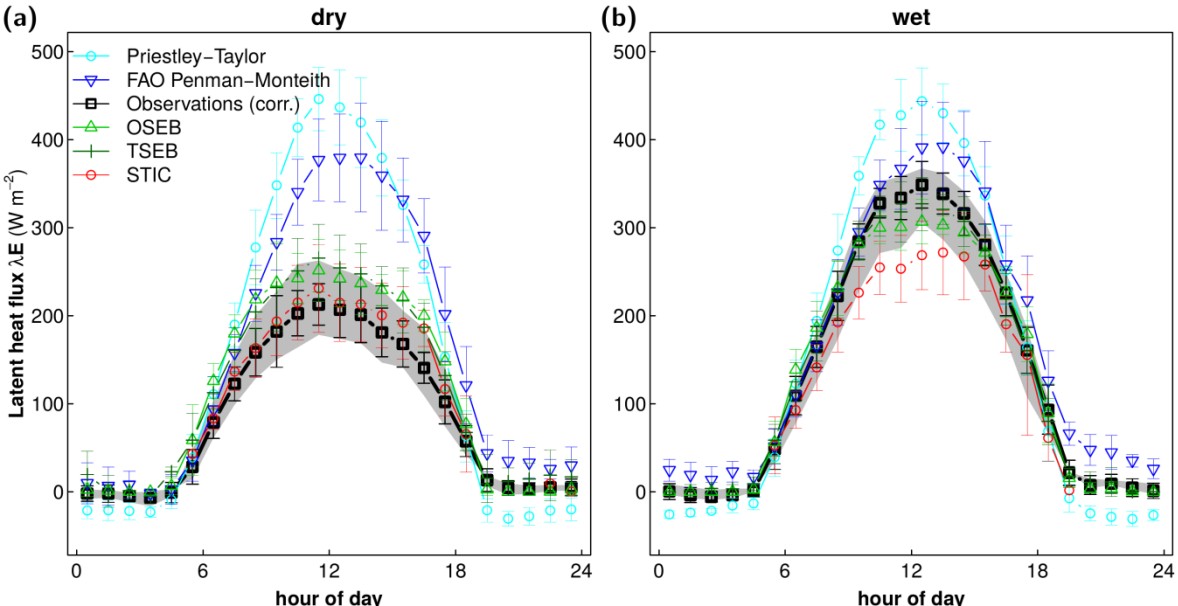

**Figure 5: Average diurnal cycle of $\lambda E$ estimates for (a) dry and (b) wet days. Error bars denote the standard deviation obtained for each hour. The bold black line with squares shows the observed latent heat flux corrected for the surface energy balance closure ($\lambda E_{\text{BRC}}$). The grey-shaded area depicts the range induced by the energy balance closure gap.**

### 3.3 Diurnal patterns of evapotranspiration

The evaluation of the diurnal cycle shows that $\lambda E$ was strongly related to the incoming solar radiation, emphasizing that $R_{\text{sd}}$ is the dominant driver of $\lambda E$ (Fig. 6). However, under wet conditions we found a marked and consistent difference between morning and afternoon in $\lambda E$ forming a CCW hysteresis loop (Fig. 6b). Using the Camuffo-Bernardi regression we found a

10 significant phase lag for the BR corrected flux ($\lambda E_{\text{BRC}}$) with a mean $t_\varphi = 15$ min under wet conditions and no significant lag under dry conditions (Fig. 7 and Table 4). The uncorrected observations showed a slightly lower wet-dry difference.

The two potential evapotranspiration estimates showed large differences in their phase lag. While the PT estimate showed a small hysteretic loop with a phase lag between $t_\varphi = 6$-9 min, the FAO-PM estimate showed a substantial loop with a phase

15 lag of $t_\varphi = 31$ min. This large phase lag of the FAO-PM estimate is very similar to the phase lag when we use a constant $g_s$ in the PM equation but with $g_{\text{av}}$ obtained from Eq. (4) using friction velocity observations (Table 4). The temperature gradient schemes (OSEB and TSEB) reproduced the observed phase lag relatively well (mean $t_\varphi = 9$ min for wet and around 0 for dry conditions). However the temperature-vapor gradient scheme (STIC) showed relatively larger phase lags under both dry and wet conditions ($t_\varphi = 14$-20 min) (Fig. 7, Table 4).



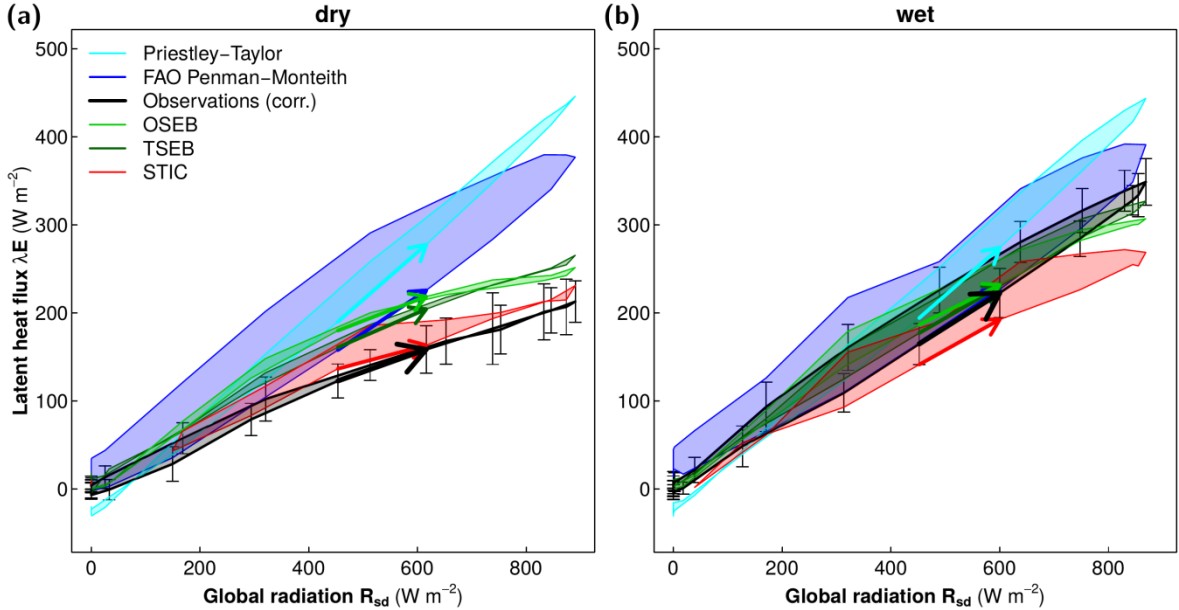

**Figure 6: Diurnal hysteresis of $\lambda E$ to $R_{sd}$ for (a) dry and (b) wet conditions of observations and different models. Bold arrows indicate the rising limb in the morning hours (7:00 to 8:00) showing a counterclockwise hysteresis of $\lambda E$ under wet conditions. Vertical arrows depict the standard deviation of $\lambda E_{BRC}$ for each hour.**

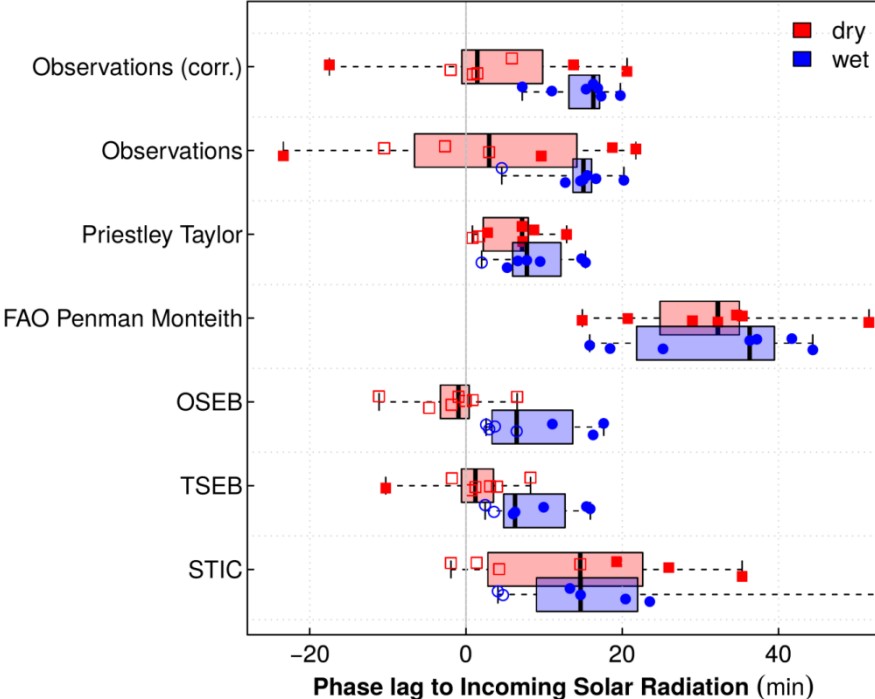

**Figure 7: Boxplot of the daily phase lag of $\lambda E$ to $R_{sd}$ for observed (without and with Bowen Ratio correction) and modeled latent heat flux using sunny, dry (red) and wet (blue) days. A positive phase lag means that $\lambda E$ follows $R_{sd}$, and thus forms a CCW hysteresis as shown in Fig. 6. Dots show the actual data for each day with filled symbols indicating significant phase lags (P < 0.05, t-test of coefficient significantly different from 0).**



### 3.4 Diurnal patterns of observed fluxes and states

In order to understand the diurnal patterns of $\lambda E$ we also analyzed the hysteresis loops of the observed surface energy balance components [$\lambda E = (R_n - G) - H$] with respect to $R_{sd}$ (Figure 8). Generally, there was only a small hysteresis in the available energy ($R_n - G$) (Table 4). The turbulent heat fluxes showed significant hysteresis under wet conditions but not under dry conditions. Interestingly, under wet conditions the CCW hysteresis of $\lambda E$ with a phase lag (mean $t_\varphi$ = 15 min) was mostly compensated by a CW hysteresis of $H$ (mean $t_\varphi$ = -22 min) (Figure 8 and Table 4). This compensation is an outcome of net available energy ($R_n - G$) showing little hysteresis for both conditions.

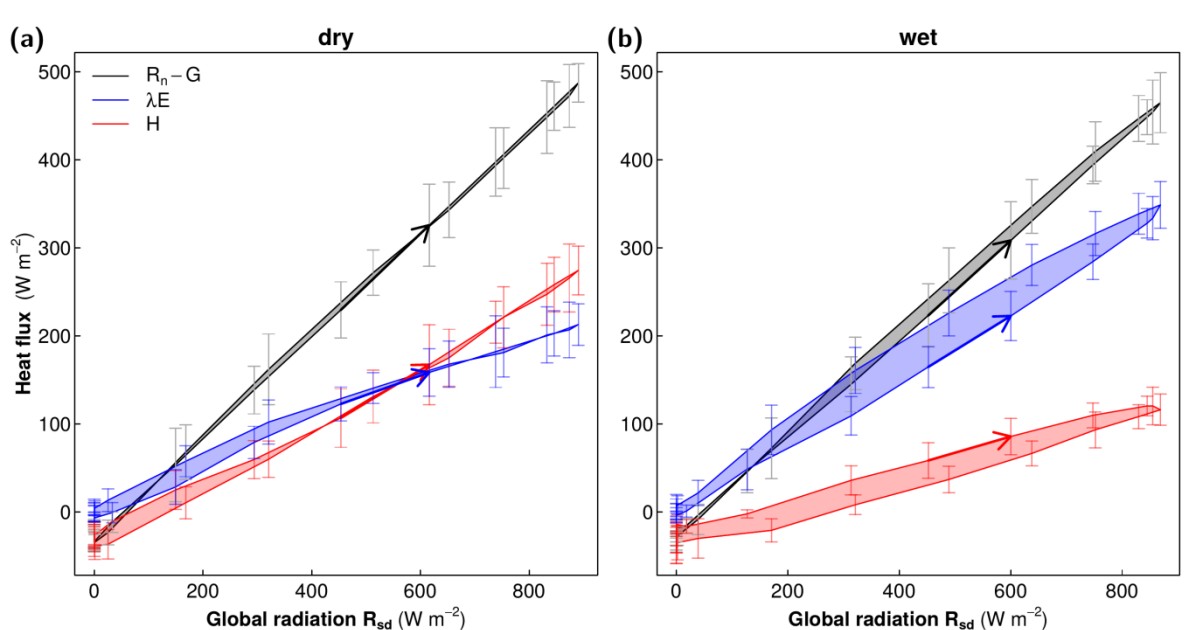

**Figure 8: Diurnal patterns of observed surface energy balance components for (a) dry and (b) wet days. The lines show the composite average and vertical bars the standard deviation for available energy (black), latent (blue) and sensible heat (red) of each hour. There is a nearly linear response of all surface heat fluxes to $R_{sd}$ under dry conditions, and a systematic hysteresis loop under wet conditions. Under wet conditions the CCW hysteresis of $\lambda E$ is mostly compensated for by a CW hysteresis of $H$.**

We next analyzed the bulk sensible heat flux formulation used in the OSEB and TSEB models to understand how the observations of temperature and the inferred aerodynamic conductances are related to each other. The diurnal patterns of both air and surface temperature revealed a strong CCW hysteresis with $R_{sd}$ (Fig. 9). Air temperature showed a more pronounced hysteretic loop than surface temperature, and with a triangular shape with higher values during the afternoon when solar radiation reduces. Interestingly, the surface-air temperature gradient, being the driving gradient for the sensible




heat flux, showed much less hysteretic behavior. The hysteresis is in a clockwise direction, with a higher gradient in the morning hours compared to the afternoon. It had a similar phase lag as $H$ (see Table 4).

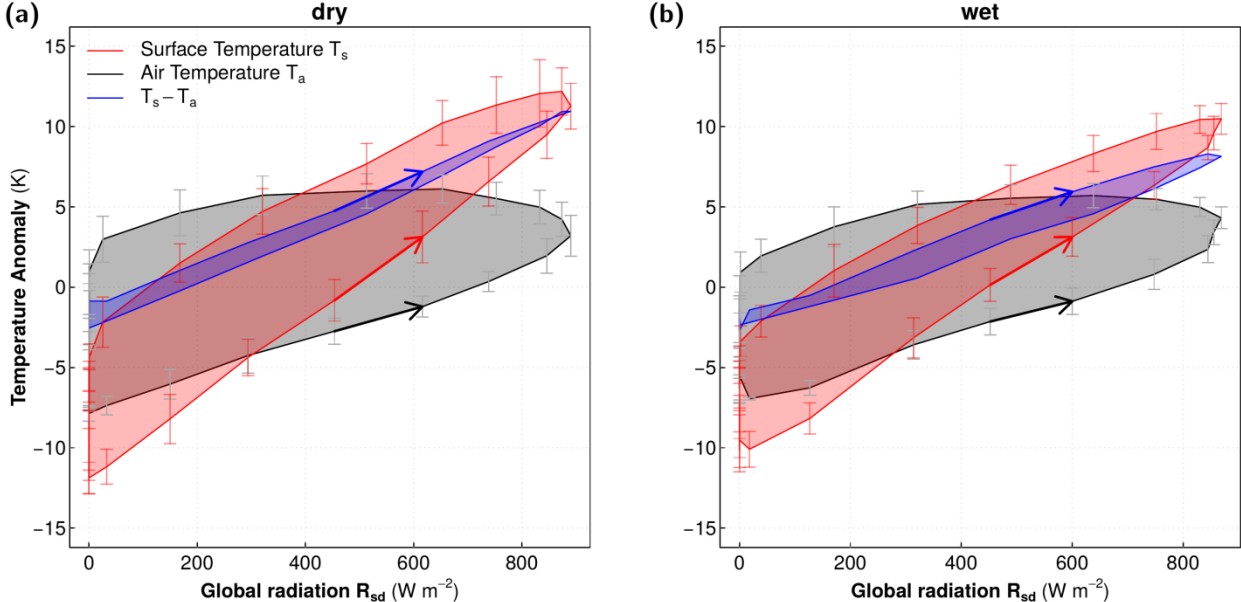

**Figure 9: Diurnal patterns of observed anomalies in surface temperature ($T_s$), air temperature at 2m ($T_a$), and their gradient ($T_s - T_a$) for (a) dry and (b) wet days. Both $T_a$ and $T_s$ show a pronounced CCW hysteresis, but the form of the hysteretic loop is significantly different, with air temperature featuring a more pronounced, triangular shape with afternoon values almost independent of incoming solar radiation. The temperature gradient, however, shows a much smaller CW hysteretic loop. Note that the temperature gradient is comparatively higher in the morning than in the afternoon, corresponding to the diurnal course of the sensible heat flux (cf. Fig. 8).**

We further analyzed different formulations of the aerodynamic conductance ($g_a$) directly inferred from measurements and from how these are represented in the models evaluated here (FAO-PM, OSEB, TSEB, STIC). We inferred the aerodynamic conductance from observations in three different ways: Firstly, we used the Eddy covariance measurements of friction velocity ($u*$) and wind speed ($u$) to estimate the aerodynamic conductance for momentum ($g_{am} = u*^2/u$). We then used the empirical formula by Thom (1972) to calculate the aerodynamic conductance for heat including the excess resistance to heat transfer (Eq. 4). Thirdly, we inferred the aerodynamic conductance from the observed sensible heat flux ($H_{BRC}$) and temperature gradient ($T_s - T_a$) by inverting $H_{BRC}$ using Eq. (5). The FAO-PM describes the aerodynamic conductance with a simple linear relationship to wind speed. OSEB and TSEB estimates the aerodynamic conductance to heat ($g_{ah}$), while STIC estimates the conductance to water vapor ($g_{av}$). Thus by comparing these different conducatance eatimates we assume similarity between the fluxes.

The different estimates for the aerodynamic conductance are compared to each other in Fig. 10 for midday conditions. Although the three observation-based estimates show some variations in the absolute value of the aerodynamic conductance,



they consistently showed a significantly greater conductance for dry days compared to wet days, suggesting a stronger aerodynamic exchange between the surface and the atmosphere under dry conditions. This difference in aerodynamic conductance is partly reproduced by the simple FAO-PM scheme which means that the median wind speed was higher under the drier conditions. The temperature-gradient schemes (OSEB and TSEB) reproduce the wet-dry difference rather well,

which also use wind speed but rely on MOST similarity and stability correction. STIC which does not use wind speed did not show any significant differences in $g_{av}$ between wet and dry conditions.

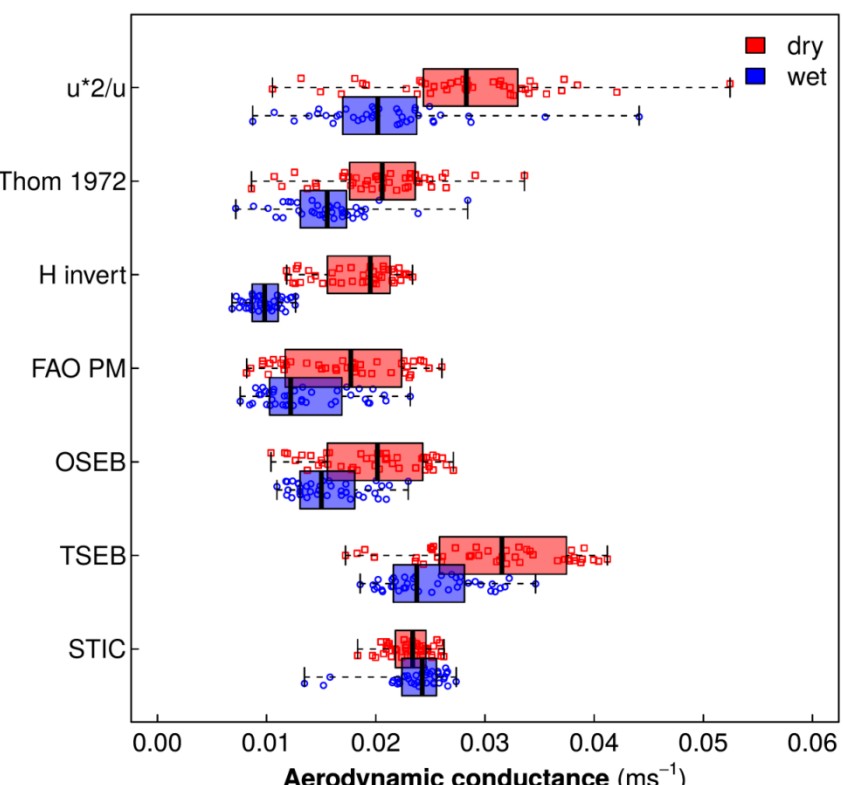

**Figure 10: Boxplot of the different estimates of aerodynamic conductance under dry (red) and wet (blue) conditions. Only sunny days are sampled and midday values (10:00-15:00) are used in the comparison. The top three estimates are directly inferred from observations, as described in the text.**

Finally we analyze the diurnal patterns of the vapor pressure deficit $D_a = e_s(T_a) - e_a$ which is a critical driver of the latent heat

flux in the PM equation. Since $D_a$ is derived from the observations, we analyzed its diurnal patterns in Fig. 11. We found that the vapor pressure in the air remained fairly constant during the day, hence it did not co-vary with $R_{sd}$, and only showed a small CW hysteresis with higher vapor pressure during the morning than during the afternoon. The saturation vapor pressure, which is a function of air temperature, however, showed a distinct and large CCW hysteresis loop with respect to $R_{sd}$, which





is consistent with the large hysteresis in air temperature (Figs. 9 and 12). As a consequence, $D_a$ also showed a distinct and large CCW hysteresis with a large phase lag of $t_\varphi =\sim 150$ min (see Table 4). This large hysteresis and phase lag is consistent with the respective characteristics of air temperature, but not with those of the temperature gradient. Furthermore, we note that the phase lag in $D_a$ did not show any significant influence of wetness, while the phase lag of the temperature gradient

5 became more negative under wet conditions (Fig. 12, Table 4). It would thus seem that the bias in PM-based estimates identified here may relate to a too pronounced role of $D_a$ in the evapotranspiration estimate.

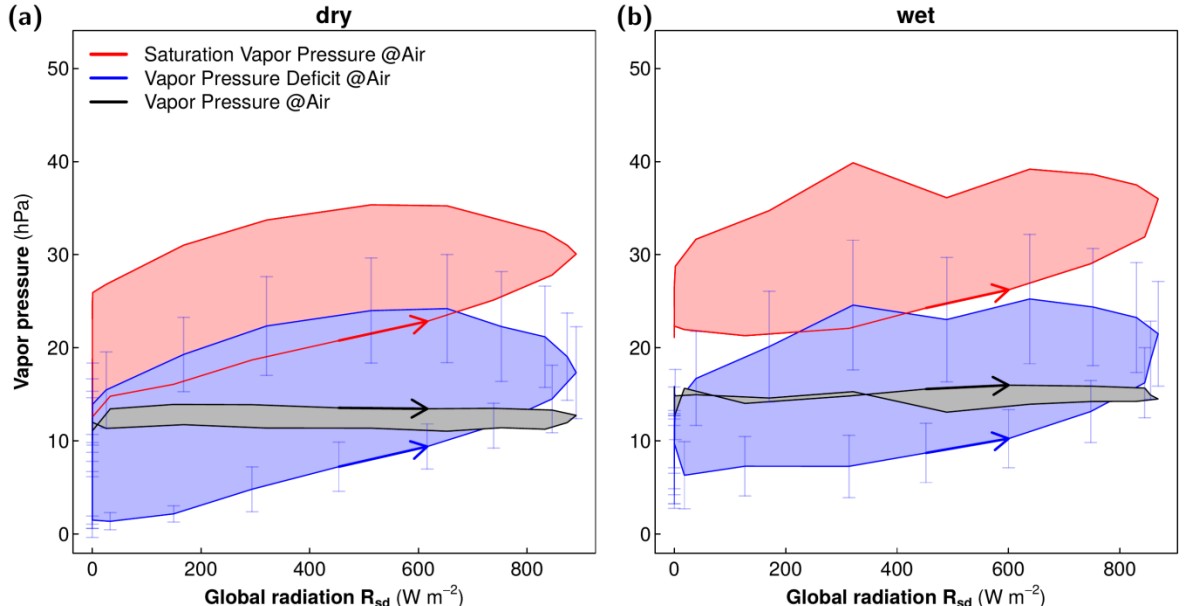

**Figure 11: Diurnal patterns of vapor pressure in air (black), the saturated vapor pressure evaluated at observed air temperature**
10 **(red), and the vapor pressure deficit (blue) for (a) dry and (b) wet days.**



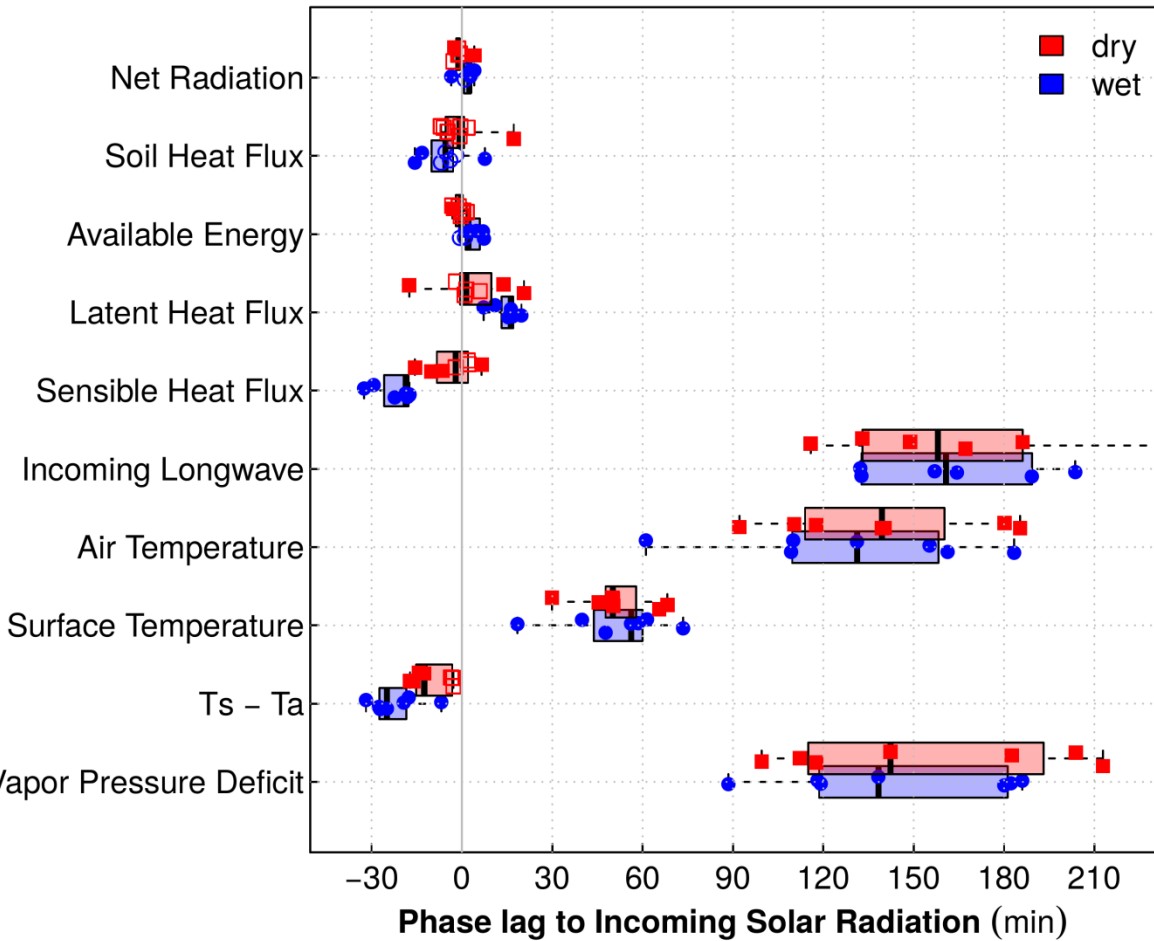

Figure 12: Phase lag to solar radiation of surface energy fluxes and surface state variables used as input for the evapotranspiration models for dry (red) and wet (blue) days. Boxplot and daily estimates with filled symbols showing significant phase lag estimates.

# 4 Discussion

## 4.1 Dominant controls of $\lambda E$ at the diurnal cycle

Our analysis of the diurnal cycle showed that $\lambda E$ follows the diurnal course of incoming solar radiation, explaining most of the variance in $\lambda E$. However, a significant non-linearity in form of a phase lag between $\lambda E$ and $R_{sd}$ was detected which showed larger $\lambda E$ for the same $R_{sd}$ in the afternoon as in the morning. We found that the lag in $\lambda E$ is accompanied by a preceding phase lag of the sensible heat flux, while the other surface energy balance components (e.g., net radiation and soil heat flux) revealed very small phase lags with $R_{sd}$. Hence, there is compensation between the phase shifts of sensible and latent heat fluxes, which becomes more apparent under the wet conditions. Our results are consistent with comprehensive




FLUXNET studies of Wilson et al. (2003) and Nelson et al. (2018) which used a different metric (median centroid) for assessing diurnal phase shifts. Wilson et al. (2003) found that $H$ precedes $\lambda E$ at most sites, with the exception of sites in a Mediterranean climate. Using the FLUXNET2015 dataset, Nelson et al. (2018) found that the median centroid of $\lambda E$ occurs predominantly in the afternoon across all plant functional types when $f_E > 0.35$, while for very dry conditions ($f_E < 0.2$) a

shift of the $\lambda E$ centroid towards the morning was found. This indicates that our results are not just applicable to Luxembourg, but are a general phenomenon which justifies a wider interpretation within temperate climates.

The obtained phase lags of the surface fluxes and variables allow for a process-based insight into the diurnal heat exchange of the surface with the atmosphere. Since there is only limited heat storage in the surface layer itself, which explains the small phase lags of the heat fluxes, the heating imbalance caused by solar radiation must be effectively redistributed. While

the soil heat flux is limited by the relatively slow heat conduction, the lower atmosphere acts as efficient heat storage. Thus, the lower atmosphere is effectively heated by surface longwave emission and the sensible heat flux, which in combination with the diurnal cycle of vertically transported turbulent kinetic energy (TKE) leads to the development of the convective planetary boundary layer (CBL) (e.g., Oke 1987). The changes in heat storage in the CBL are reflected by the very large phase lags for air temperature and longwave downwelling radiation, which both have a phase lag of about 2.5h. This large

phase lag of air temperature then shapes (i) the vertical surface-air temperature gradient, which drives the sensible heat flux, and (ii) the vapor pressure deficit of the air. Despite the complexity of processes within the convective boundary layer, including the morning transition and entrainment at its top, we find that all surface energy components correlate strongly with solar radiation (Table 4). What this suggests is that the state of the surface-atmosphere system is predominantly shaped by fluxes, particularly by solar radiation as its primary driver, with the state in terms of temperatures and humidity gradients

adjusting to these fluxes, rather than the reverse, where the state (in terms of temperature and humidity) drives the fluxes.

These interactions are also affected by soil water availability, as reflected in the phase lags. This is most clearly seen for the surface-air temperature gradient and the sensible heat flux, whose phase lag is two times larger for wet than for dry days. This means that for the same solar radiation forcing we find higher values of the sensible heat flux in the morning than in the

afternoon. The effect of water availability is also seen for the phase lag of the latent heat flux and to a lesser extent for the soil heat flux. These findings agree well with studies which use the diurnal centroid method, showing that moisture limitation decreases the lag in timing of $\lambda E$ (Wilson et al. 2003, Xiang et al., 2017, Nelson et al., 2018). The phase shift of $D_a$ might enhance evaporation at the cost of the sensible heat flux during the afternoon under sufficient moisture availability. However, under drier conditions, our findings suggest that the surface heats more strongly and generates more buoyancy,

which is reflected by higher aerodynamic conductances as compared to the wet conditions (Fig. 10). The larger aerodynamic conductance would then enable a more effective sensible heat exchange, and would thus lower the phase difference between the sensible and latent heat fluxes.





Note, that our interpretation disregards the effects of horizontal advection of moisture and temperature. Events of strong advection, e.g. of temperature can add heat to the surface energy balance and thus alter the diurnal cycle. Similarly, events of dry air advection may enhance local $\lambda E$ at the cost of the sensible heat flux. Since we used composite averages and statistics over a set of days we aim to reduce the impact of such advective events. We expect that it is unlikely that such events

occurred throughout all wet / dry days in a consistent manner.

## 4.2 Using phase lags to identify model biases

Our comparison of different modeling approaches shows that by using phase lags, one can identify biases in evapotranspiration parameterizations and relate these towards processes for a better understanding of surface-atmosphere

interactions under different conditions of water availability. One of our main findings is that the surface energy balance fluxes and the temperature gradient have a comparatively small phase lag to the incoming solar radiation, while air temperature and vapor pressure deficit have substantial phase lags. This difference in phase lags can then be used to infer biases in estimates of evapotranspiration. In our application of this approach to observations of one site in a temperate climate we found that evapotranspiration exhibits a comparatively small phase lag, indicating that it was dominantly driven

by solar radiation and temperature gradients, and not by air temperature and the water vapor pressure deficit. This interpretation is consistent with studies of non-water-stressed evapotranspiration that is best represented by potential evapotranspiration schemes which are primarily driven by net radiation, as demonstrated for FLUXNET observations by Maes et al., (2018) and for climate model simulations by Milly and Dunne, (2016). Milly and Dunne (2016) interpreted these findings in terms of strong feedbacks between the surface and the atmosphere, which couple the surface variables and

which result in a top-down energy constraint that is well captured by energy-only formulations.

Our analysis allows to better understand the relevance of the feedbacks which occur at a sub-daily time scale. These feedbacks are driven by the redistribution of heat gained by absorption of solar radiation at the surface, which causes a significant co-variation of the input variables to incoming solar radiation (Table 4). This is especially important for the vapor

pressure deficit of the air which acts as a driver of $\lambda E$ and is also known to affect the stomatal conductance (Jarvis 1976, Jarvis and McNaughton 1986). De Bruin and Holtslag (1982) showed that a positive correlation between $D_a$ and $R_n$ allows simplifying the complex PM equation to a form similar to equilibrium evaporation (Eq. 7) with net radiation as the dominant driver. Therefore, simpler, energy based formulations for $\lambda E$ show similar performance as PM based approaches, but with less input parameters (De Bruin and Holtslag 1982, Beljaars and Bosveld 1997). The challenge of the PM equation is then a

parameterization of the conductances, which must capture the feedbacks included in the input data. Since the co-variation originates from the diurnal redistribution of heat, a mismatch would then clearly be seen at the diurnal timescale. Hence by focusing on the internal relationship of the modeled $\lambda E$ to $R_{sd}$ at the diurnal time scale we found that (i) the Penman-Monteith based approaches showed a consistently larger phase lag than what was actually observed and (ii) these approaches did not



show a reduction of the phase lag under dry conditions. The PM approaches use the vapor pressure deficit as input variable which showed a substantial hysteresis loop in the order of 2.5 h lagging $R_{sd}$. This is due to the temperature dependency of the saturation vapor pressure, while the actual vapor pressure shows no relationship with $R_{sd}$. Besides $D_a$, all other input variables to the Penman-Monteith approaches used here (both FAO and STIC) showed minor phase lags with respect to $R_{sd}$.

Since the surface conductance in FAO Penman-Monteith is fixed with time, the resulting prediction of potential $\lambda E$ showed a significant and large phase lag in the order of 0.5 h. Even when we use the observed aerodynamic conductance as input, the effect remains the same, which emphasizes that a constant surface conductance results in biases in the diurnal cycle of $\lambda E$. In contrast to assuming a constant $g_s$, STIC computes $\lambda E$ through analytical estimation of $g_s$ and $g_{av}$ from the information of both, the surface-air temperature gradient and the vapor pressure deficit. Given that the lag of $D_a$ to $R_{sd}$ was similar for both

dry and wet conditions (Fig. 12), $\lambda E$ from STIC also revealed a similar pattern and there was no substantial difference in phase lag of $\lambda E$ from STIC between the wet and dry conditions. Hence, our analysis indicates that the PM-based approaches used here overestimated the effect of water vapor deficit on actual evapotranspiration, which, in the end, reflects in the estimation of the surface and the aerodynamic conductance to water vapor.

The temperature-gradient approaches used here (OSEB and TSEB) are structurally different from the PM approaches, since they infer $\lambda E$ from the residual of the surface energy balance and thus do not explicitly deal with the aerodynamic and surface conductance of water vapor. The phase lag analysis of the environmental variables used to drive the predictive models of $\lambda E$ helped to identify an important benefit of the temperature-gradient approaches over the Penman-Monteith based approach.

The temperature-gradient approaches employ the vertical temperature gradient ($T_s - T_a$) which showed a significant counter-clockwise, i.e. a leading hysteretic loop, which is in the order of the phase shift detected for the sensible heat flux (Fig. 12). In addition, there is a distinct and significant increase of the phase shift in both the temperature gradient and the sensible heat flux under the wet conditions. Hence, the temperature gradient as input contains valuable information on water limitation in terms of the magnitude (i.e. the slope of ($T_s - T_a$) to $R_{sd}$) and the diurnal phase lag (see Table 4).

While the PM approaches must identify two conductances simultaneously, the temperature-gradient approaches only need to parameterize the aerodynamic conductance to heat ($g_{ah}$) using wind-speed as input. Thereby we found that these approaches agreed well with the approximated $g_{ah}$ from the EC tower, which shows an enhanced conductance under dry conditions. Contrarily, the diagnosed $g_{av}$ from STIC did not show substantial differences between dry and wet conditions, pointing to the

difficulty of the analytical approach and its associated assumptions to identify two bulk conductances parameters from the available radiometric and meteorological data (Mallick et al., 2018) for the climatic conditions in which these were evaluated here.





Note, that we evaluated a temperate grassland site which experienced an exceptional summer drought. Thereby, the evaporative fraction did not decline below 0.3. In semi-arid ecosystems the evaporative fraction may decrease substantially below 0.3 and Nelson et al., 2018 showed that there is a morning shift of $\lambda E$ (analogous to a negative phase lag) under very dry conditions ($f_E < 0.2$). This points towards a different stomatal regulation changing the diurnal course of surface

conductance. While it was shown by Bhattarai et al., (2018) that STIC performs well also under semi-arid conditions, temperature-gradient approaches can show larger biases under semi-arid conditions (Morillas et al., 2013). The difficulty of temperature-gradient approaches are predominantly in the parameterization of aerodynamic conductance of heat which becomes more challenging under these very dry conditions (Kustas et al., 2016).

The relevance of the diurnal time scale for the problem of surface conductance parameterizations was already highlighted by Matheny et al., 2014. However, they and others evaluated the diurnal patterns of the hysteretic loops between $\lambda E$ and $D_a$ (see also Zhang et al., 2014, Zheng et al., 2014). Given that solar radiation is the cause of the strong L-A feedbacks at the diurnal time scale we believe that solar radiation is better suited as a reference variable than $D_a$. Our results show that the new metric of the phase lag of heat fluxes and surface states to incoming solar radiation reveals important biases of evapotranspiration

schemes often used in remote sensing. These biases may well be compensated for at a longer time scales (Matheny et al., 2014) but would lead to biased sensitivities with respect to climate change (Milly and Dunne 2016). Here, we applied the phase lag metric to observationally driven evapotranspiration schemes. In the future, we plan to apply these new metrics based on hysteretic loops to model outputs of land-surface models (such as NOAH-MP, Niu et al., 2011) as well as of fully coupled surface-atmosphere simulations in order to detect and to identify errors in the parameterization of state-of-the-art

LSMs.

## 5 Conclusions

We quantified the non-linear relationship of evapotranspiration to incoming solar radiation in terms of a phase lag at the diurnal time scale. Our findings have practical implications for remote sensing based $\lambda E$ retrievals and for land-surface model evaluation and calibration. We evaluated three structurally different schemes which are used in remote sensing based

applications for a temperate grass site which experienced a summer drought. Our results showed that temperature gradient schemes show a higher correlation and thus better represent the diurnal cycle than a combined temperature-vapor gradient scheme. What this means is that for our site, the temperature gradient between the surface and the near-surface atmosphere contained more important information of water limitation than the water vapor pressure deficit (which is also difficult to retrieve from remote sensing (Kalma et al., 2008, Yang et al., 2015)). Furthermore, schemes which use vapor pressure deficit

as additional input (such as the Penman-Monteith formulation), require a dynamic, i.e. time dependent characterization of surface conductance to account for the strong phase lag in vapor pressure deficit. Hence, our results suggest that simpler $\lambda E$ approaches based on the surface energy balance and surface temperature may be more suitable to estimate evapotranspiration





from observational data (e.g. remote sensing data) in climates without substantial water stress. It would seem that these surface observations already contain the imprint of land-atmosphere interactions, whereas in the case of coupled land surface-atmosphere models these interactions are explicitly resolved. Hence, detailed models of aerodynamic conductance and its interaction with the environment are of crucial importance for skillful climate predictions including the carbon cycle

(Prentice et al., 2014, Wolf et al., 2016; Konings et al., 2017).

We suggest that an evaluation of these schemes should be based on the diurnal time scale, because a land-atmosphere exchange scheme must accomplish a balance between the surface energy balance with small imprints of heat storage changes and the lower atmosphere with strong imprints of heat storage changes (Kleidon and Renner 2017). Although a mismatch of

the diurnal patterns may not be seen at the aggregated time scales of days and months, it may lead to biased model sensitivities (Matheny et al., 2014) which would be crucial for assessing climate change impacts (Milly and Dunne 2016). Here, we analyzed observationally driven evapotranspiration schemes and their inputs, which revealed an apparent energy constraint. This constraint, which appears as strong correlation of surface fluxes and gradients to incoming solar radiation should be correctly represented by any land-surface model which resolves the land-atmosphere interaction. While this may

sound trivial, recent benchmarking studies showed that current state-of-the-art land-surface models have difficulties to represent the strong link of turbulent heat fluxes to solar radiation (Best et al., 2015; Haughton et al., 2016). Our findings provide an explanation of why this is so difficult to achieve and we suggest that further information is gained by evaluating land surface schemes in terms of phase lags in surface fluxes and states such as the sensible and soil heat flux including the diurnal dynamics of surface and air temperature. Correctly representing these metrics will lead towards a more accurate

representation of the diurnal heat and mass exchange of the land with the atmosphere.

## 6 Code availability

Code to perform OSEB and TSEB simulations can be found at

https://github.com/ClaireBrenner/pyTSEB_Renner_et_al_2018. Code to simulate STIC1.2 simulations is available upon request at Kaniska Mallick (LIST, kaniska.mallick@list.lu). Code to perform data analysis and figures can be obtained from the main author upon request.

## 7 Data availability

Data can be obtained upon request from the main author. Observational data from the EC site can be obtained from Hans-

Dieter Wizemann, hans-dieter.wizemann@uni-hohenheim.de.



## 8 Author contribution

MR and AK conceived the analysis of the diurnal cycle. VW, KS and IT designed the field campaign. HW carried out the EC measurements and EC data processing. CB performed OSEB/TSEB simulations. KM performed STIC1.2 simulations. MR merged the data and performed data analysis. MR and LC developed the phase lag computation. JW provided ancillary simulation data. IT provided climate information. MR prepared the manuscript with contributions from all co-authors.

## 9 Competing interests

The authors declare that they have no conflict of interest.

## 10 Special issue statement

This draft shall contribute to the upcoming Joint special issue in HESS (lead journal) and ESSD:

Linking landscape organisation and hydrological functioning: from hypotheses and observations to concepts, models and understanding

## 11 Acknowledgements

This study was supported by the German Research Foundation (DFG) through funding of the research unit "From Catchments as Organised Systems to Models based on Dynamic Functional Units – CAOS" (FOR 1598) within the sub-project "Understanding and characterizing land surface-atmosphere exchange and feedbacks" (Project number 182331427). MR and KL were funded by DFG grant number (KL 2168/2-1). CB was supported by the Austrian Science Fund (FWF) through funding of the CAOS Research Unit (I 2142-N29). KM was supported by the Luxembourg Institute of Science and Technology (LIST) through the project BIOTRANS (grant number 00001145), CAOS-2 project grant (INTER/DFG/14/02) funded by FNR (Fonds National de la Recherche)–DFG, and HiWET consortium funded by the Belgian Science Policy (BELSPO) – FNR under the programme STEREOIII (INTER/STEREOIII/13/03/HiWET; CONTRACT NR SR/00/301). IT was supported by the Luxembourg Institute of Science and Technology (LIST) through the project BIOTRANS (grant number 00001145).

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



**Table 1: Variables provided by the surface energy balance station and used for this work.**

| Variable | Symbol | Unit |
|---|---|---|
| **Horizontal wind components** | $u, V$ | m s$^{-1}$ |
| **Vertical wind** | $w$ | m s$^{-1}$ |
| **Sensible heat flux** | $H$ | W m$^{-2}$ |
| **Latent heat flux** | $\lambda E$ | W m$^{-2}$ |
| **Ground heat flux** | $G$ | W m$^{-2}$ |
| **Upward shortwave radiation** | $R_{su}$ | W m$^{-2}$ |
| **Incoming shortwave radiation** | $R_{sd}$ | W m$^{-2}$ |
| **Surface shortwave albedo** | $A$ | |
| **Upward longwave radiation** | $R_{lu}$ | W m$^{-2}$ |
| **Downward longwave radiation** | $R_{ld}$ | W m$^{-2}$ |
| **Friction velocity** | $u*$ | m s$^{-1}$ |
| **Air temperature** | $T_a$ | K, °C |
| **Relative humidity** | $r_H$ | % |
| **Surface air pressure** | $p$ | hPa |
| **Precipitation** | $P$ | mm |
| **Soil moisture (5, 15 and 30 cm)** | $\theta$ | v/v |
| **Soil temperature (5, 15 and 30 cm)** | $T_{soil}$ | K |

**Table 2: Input variables used in the different evapotranspiration schemes.**

| Scheme | $R_n$ | $G$ | $T_a$ | $T_s$ | $e_a$ | $e_s$ | $u$ | Other parameters |
|---|---|---|---|---|---|---|---|---|
| **Priestley-Taylor** | Obs | Obs | Obs | | | | | |
| **Penman-Monteith (with constant $g_s$)** | Obs | Obs | Obs | | Obs | Obs | Obs | $g_{ah,Thom} = $ f$(u,u*)$ (3), $g_s = $ const |
| **FAO Penman-Monteith** | Obs | Obs | Obs | | Obs | Obs | Obs | $g_{av} = u/208$, $g_s = 1/70$ m/s |
| **OSEB** | Obs | Obs | Obs | Obs | | | Obs | $h_c$ |
| **TSEB** | Obs | Obs | Obs | Obs | | | Obs | $f_c, f_g, h_c$ |
| **STIC** | Obs | Obs | Obs | Obs | Obs | Obs | | |





**Table 3: Statistics for all days, and sunny wet or dry days based on 30min values during daytime hours 6:00-18:00. Performance statistics, root mean square error (rmse) and explained variance r$^2$ are computed with respect to the observed latent heat flux corrected for the closure gap by the Bowen Ratio ($\lambda E_{BRC}$). As a reference we also provide statistics for the uncorrected, observed latent heat flux ($\lambda E_{uncor}$). Potential evapotranspiration estimates are Priestley-Taylor (PT) and FAO Penman-Monteith (FAO-PM) reference evapotranspiration. Actual $\lambda E$ estimates are provided by the three schemes. Statistics are computed for all days and for clear-sky days classified as "wet" and "dry".**

| Statistic | Period | $\lambda E_{BRC}$ | $\lambda E_{uncor}$ | PT | FAO-PM | OSEB | TSEB | STIC |
|---|---|---|---|---|---|---|---|---|
| **mean** | all | 178 | 145 | 259 | 225 | 202 | 204 | 170 |
| **mean** | wet | 264 | 213 | 325 | 295 | 255 | 259 | 218 |
| **mean** | dry | 164 | 134 | 315 | 286 | 212 | 209 | 180 |
| **rmse** | all | 0 | 40 | 106 | 82 | 41 | 43 | 46 |
| **rmse** | wet | 0 | 57 | 71 | 52 | 29 | 24 | 66 |
| **rmse** | dry | 0 | 33 | 169 | 141 | 57 | 58 | 45 |
| r$^2$ | all | 1.00 | 0.94 | 0.72 | 0.62 | 0.85 | 0.84 | 0.72 |
| r$^2$ | wet | 1.00 | 0.91 | 0.96 | 0.83 | 0.92 | 0.92 | 0.66 |
| r$^2$ | dry | 1.00 | 0.93 | 0.75 | 0.55 | 0.62 | 0.61 | 0.44 |

**Table 4: Results of the Camuffo-Bernardi regression model with mean (standard deviation) for wet and dry days. The slope of the variable against $R_{sd}$ is represented by $b$ (note that the unit of b depends on the unit of the variable) and the phase lag to incoming solar radiation is converted to minutes. The adjusted explained variance by the multi-linear regression model is given in column R$^2$.**

| Variable | Moisture conditions | Slope $b$ | Phase Lag (in min.) | Adjusted R$^2$ |
|---|---|---|---|---|
| Net Radiation | wet | 0.7162 (0.0106) | 1 (3) | 0.998 |
| | dry | 0.6980 (0.0119) | -1 (2) | 0.998 |
| Soil Heat Flux | wet | 0.1483 (0.0194) | -6 (8) | 0.964 |
| | dry | 0.1261 (0.0173) | -0 (8) | 0.968 |
| Available Energy | wet | 0.5679 (0.0122) | 3 (3) | 0.998 |
| | dry | 0.5719 (0.0180) | -1 (2) | 0.998 |
| Sensible Heat Flux | wet | 0.1715 (0.0275) | -22 (6) | 0.964 |
| | dry | 0.3388 (0.0470) | -3 (8) | 0.988 |
| Incoming Longwave | wet | 0.0340 (0.0092) | 133 (84) | 0.600 |
| | dry | 0.0263 (0.0115) | 176 (51) | 0.459 |
| $\lambda E_{BRC}$ | wet | 0.3992 (0.0186) | 15 (4) | 0.990 |
| | dry | 0.2380 (0.0317) | 3 (12) | 0.981 |
| $\lambda E_{uncor}$ | wet | 0.3284 (0.0289) | 14 (5) | 0.967 |
| | dry | 0.1939 (0.0271) | 2 (16) | 0.963 |





| | | | | |
|---|---|---|---|---|
| Priestley Taylor | wet | 0.5352 (0.0278) | 9 (5) | 0.997 |
| | dry | 0.5237 (0.0412) | 6 (4) | 0.996 |
| Penman Monteith const. $g_s$ | wet | 0.3995 (0.0354) | 31 (8) | 0.982 |
| | dry | 0.3888 (0.0404) | 35 (11) | 0.974 |
| FAO Penman Monteith | wet | 0.4241 (0.0435) | 31 (11) | 0.980 |
| | dry | 0.4211 (0.0537) | 31 (12) | 0.981 |
| $\lambda E$ OSEB | wet | 0.3718 (0.0100) | 9 (6) | 0.976 |
| | dry | 0.2978 (0.0372) | -2 (5) | 0.948 |
| $\lambda E$ TSEB | wet | 0.3793 (0.0228) | 9 (5) | 0.989 |
| | dry | 0.2843 (0.0545) | 1 (6) | 0.962 |
| $\lambda E$ STIC | wet | 0.3037 (0.0695) | 20 (19) | 0.876 |
| | dry | 0.2387 (0.0655) | 14 (14) | 0.892 |
| Air Temperature | wet | 0.0088 (0.0008) | 130 (41) | 0.742 |
| | dry | 0.0084 (0.0017) | 138 (35) | 0.685 |
| Skin Temperature | wet | 0.0203 (0.0010) | 51 (18) | 0.923 |
| | dry | 0.0228 (0.0027) | 51 (13) | 0.933 |
| $T_s - T_a$ | wet | 0.0116 (0.0013) | -22 (8) | 0.966 |
| | dry | 0.0145 (0.0017) | -10 (7) | 0.973 |
| Vapor Pressure | wet | 0.0003 (0.0015) | 125 (188) | 0.267 |
| | dry | -0.0003 (0.0012) | 52 (247) | 0.316 |
| Vapor Pressure Deficit | wet | 0.0371 (0.0059) | 53 (15) | 0.961 |
| | dry | 0.0409 (0.0071) | 57 (19) | 0.953 |