# Peer review of "Understanding model biases in the diurnal cycle of evapotranspiration: a case study in Luxembourg"

_Hydrology and Earth System Sciences, 2018_

## Referee Comment (RC1) · Anonymous Referee #1 · 20 Aug 2018

The study attempts to assess the model biases in the diurnal cycles of evapotranspiration and to analyze the influence of observed input variables under dry and wet conditions. Much effort has been undertaken to analyze a wealth of observed and modeled data. The approach applied in this study is relatively logical. The findings in the paper may be rational. Therefore, I appreciate the authors' effort to handle such much work. However, I have one concern on the presentation. The paper would be publishable in HESS after minor revisions if the author satisfactorily address my concerns.

Lines 22-30 in page 1, This part should be simplified and keep concise for good readability.

Lines 20-23 in page 2, This study focus on revealing the model biases of evapotranspiration by multivariate metrics, therefore recent literatures should be summarized such

as Zhou et al., (2018, published in ACP, doi: 10.5194/acp-18-8113-2018) and Zhou et al., (2017, published in JC, doi: 10.1175/JCLI-D-16-0903.1) that investigated the model biases of regional warming in current reanalysis products and attributed those to the modeled land-atmosphere energy budgets and precipitation frequency.

Section Introduction in pages 2-4, some recent relevant literatures should be summarized in the paper, such as van Heerwaarden et al., (2010, published in JC, doi: 10.1175/2010JHM1272.1).

Lines 11-21 in page 5, There are other approaches to regress this type of the response. Some reasons of the selection of the Camuffo-Bernardi equation should be provided for good readability.

Lines 25-end in page 8, The average gap is up to 67 Wm-2 and then the diurnal cycle may has a larger gap. How to quantify the influence of energy balance closure gap (before and after correction) on the magnitude and phase lag in the paper?

Line 21 in page 22, how to justify the sentence ('These interactions are also affected by soil water availability, as reflected in the phase lags.') in the paper? whether adding related literatures or not?

Section Conclusions in pages 25-26, The author should rewrite this part to make its logic smooth. If necessary, some discussion should be added to help the readers understand the importance and advantages of this study.
* * *

---

## Referee Comment (RC2) · Anonymous Referee #2 · 29 Aug 2018

General Comments

The author's objectives of the study were to use measurements of hourly incoming shortwave radiation as an independent forcing of the land-atmosphere exchange and assess the response and phase lags of surface heat fluxes. The authors argue that models of ET should be able to capture the magnitude of hysteretic loops under different conditions.

The writing is good, and the article is well structured. The major concern I have is that incoming solar radiation (Rsd) is used rather than the available energy (Rn-G). It is expected that phase lags would occur between Rsd and LE since much energy is stored in the ground surface during the day and then released at night, so it is unclear what the novel aspects of the paper really are. All the results are fairly straightforward,

but again, they are to be expected based on the study design of using Rsd instead of Rn-G. Additionally, descriptions of what was assumed or used as input to the models, (specifically the PT and FAO-56 PM equations) is not adequate, only Rn is in the PT equation listed, not Rn-G as stated in the original equation, so there could be an error in the analysis. It is unclear if measured G was used in the FAO-56 PM equation, or if it was estimated, and same goes for Rn. Based on the lack of clarity as to what Rn and G model was used in the FAO-56 approach, those results cannot be assessed as is. While there is some good discussion on process, the novelty of the study is lacking. Perhaps a more useful and/or complementary analysis would be to focus on the hourly distribution of the energy balance closure ratio, and assess the controlling factors of the distribution, if any, as it relates to soil moisture and other conditions. As the manuscript is now, unfortunately, I recommend rejection with an opportunity to re-submit once the study design and novel aspects of the study are reconsidered.

Specific Comments Pg 2 Line 21-22: Would be good to give a quick summary of these metrics, and why some are more useful than others if they are to be used or referenced later. This would be good so that when the alternative metric is proposed below the reader has some context.

Pg 2 Line 29-30: LE is strongly correlated with Rsd, not the other way.

Pg 3 Line 2: I don't think that the other controls (other than Rn and Rsd) on LE remains unclear... it is pretty simple from an energy balance perspective (which is what is being discussed so far in terms of Rn and Rs) ... LE = Rn – H – G ... Lots to dig into with H obviously... and G, and perhaps that is where some of the controls need more study?

Page 3 Line 16: Why is Rn not used? Better yet, why isn't Rn-G (available energy) used? I don't see that why Rn (and Rn-G) is not used if the authors are indeed trying to better understand controls on LE... longwave radiation is a big component of Rn, and G lags no doubt control some of this hysteresis. I feel that the authors are missing too much energy if they just focus on Rsd.

[Figure]

Page 3 Line 23: The PT equation requires Rn, not Rsd, so how can you say you focus on using Rsd, but use the PT equation and not use Rn? Same with the PM equation. Please explain how and if Rs is used in these equations here and you can go into greater detail in the methods if needed.

Page 4 Line 7: but RH and VPD is coming from gridded weather data, no? So this is a forcing and outside the evaporation model, correct?

Page 6 Line 28-34: This is concerning since the heat storage in the soil slab above the G plate was estimated rather than measured. Sounds like the estimate didn't consider changes in soil moisture, which is a big factor in the potential to store heat within soils. Any errors in the estimate, or bad heat storage measurements could cause "perceived hysteresis" when comparing to other energy balance components. When was the harmonic calibrated, to dry or wet conditions, or both? Did the harmonic behave differently (have different parameters) when assessed during wet vs dry conditions as anticipated?

Page 8 Line 31: What time step was Qgap (the energy balance closure) assessed? Every 30min?

Page 9 Line 1-5: Was this done at 30min time steps?

Page 10 Line 25: The PT equations uses Rn-G, not simply Rn as written. What did you use? Rn or Rn-G (see equation 14 of the PT paper - ftp://ftp.library.noaa.gov/docs.lib/htdocs/rescue/mwr/100/mwr-100-02-0081.pdf ). The phase lag results can't be interpreted until this is cleared up.

Page 11 Line 14-16: Is Rsd used and then Rn and G is estimated following the procedures of FAO-56, or is the measured Rn-G used? This needs to be spelled out to understand the results.

Figure 6. Be consistent calling incoming shortwave Rsd vs Global radiation. . . you say both.

Figure 7 isn't very useful since it is Rsd on the x, and not Rn-G. I guess I don't see the point since phase lag is to be expected (and greater for wet conditions as shown), and it is unclear how G was considered in the FAO approach.

Page 17 line 4-5: The authors state that "Generally, there was only a small hysteresis in the available energy (Rn – G ) (Table 4)" which is exactly what one would expect if Rn-G was used. So by not including longwave and G there is phase lag, which is to be expected, so I don't see the point of the paper really... Also, there would be more phase lag in wet soil conditions, than in dry conditions since heat storage is greater when there is more water in the soil. By not considering G, you get phase lags... is there something novel to see here?

Page 20 and Figure 11: The results of the hysteresis in humidity variables are what you would expect. The VPD is lowest in the morning, and highest in the mid to late afternoon, and largely a function of es, since it is a fairly humid environment, so what is novel here?

Page 25 Line 22: Yes this was quantified, but it was expected, and it changes in time and space, based on the land surface conditions, and met. forcings.

Page 25 Line 23: Explain exactly how these results have practical application for remote sensing based models? This was never fully described, that is why this phase lag issue is so important for remote sensing studies of LE to consider or include.

Page 25 Line 30-33: There is too little information on the specifics in the paper of FAO-PM approach applied to assess if this is a correct conclusion.

---

## Author Comment (AC1) · 31 Aug 2018

The reviewer claimed three critical shortcomings of the manuscript that we think are not shortcomings, but rather misunderstandings. This is why we want to address them immediately to avoid further misunderstanding. A more detailed response is posted at a later time.

**1   Input to models**

**Reviewer2: "descriptions of what was assumed or used as input to the models, (specifically the PT and FAO-56 PM equations) is not adequate ... Based on the**

**lack of clarity as to what $R_n$ and $G$ model was used in the FAO-56 approach, those results cannot be assessed as is."**

Reply: All models have been driven by the observational data which is important because this allows a fair comparison between models. A list of input is provided in Table 2 of the manuscript.

Specifically, both potential evapotranspiration estimates, the Priestley-Taylor evapotranspiration and FAO Penman-Monteith estimate, use available energy ($R_n - G$) as input. We apologize that the soil heat flux was missing in the Priestley-Taylor Equation (Eq. 7) and will correct this typo in the revision. The calculations are, however, not affected by this typo. Also, the FAO Penman-Monteith equation was driven with net radiation and soil heat flux from the observations and not from one of the empirical replacements as provided in the FAO-56. Hence we can assure that the findings of systematically different diurnal cycles of the Penman-Monteith driven models is indeed related to the model formulation and not to errors in the analysis.

All code (and data) to reproduce the analysis will be provided in a public accessible repository with the revision of the manuscript.

**2   Solar radiation vs. Available Energy ($R_n - G$)**

**Reviewer2: "The major concern I have is that incoming solar radiation ($R_{sd}$) is used rather than the available energy ($R_n - G$)."   "I don't see that why $R_n$ (and $R_n - G$) is not used if the authors are indeed trying to better understand controls on LE"**

Reply: Our main reasoning is that available energy is not an independent variable as it depends on surface temperature. We specifically choose incoming Solar radiation (= Global Radiation) ($R_{sd}$) as the reference for the phase shift analysis. While the

term available energy ($R_n$ – G) is often used as input to evapotranspiration schemes, it is important to remind that the net radiation $R_n$ is the radiation budget comprised of shortwave and longwave components:

$$R_n = R_{sd} - R_{su} + R_{ld} - R_{lu} \qquad (1)$$

With upwelling shortwave radiation ($R_{su}$), downwelling longwave radiation $R_{ld}$ and upwelling longwave radiation $R_{lu}$. $R_{lu}$ is strongly related to skin temperature and cannot be regarded as an independent variable. Therefore, the radiation budget ($R_n - G$) cannot be regarded as independent from the surface heat fluxes (see e.g. Ohmura (2014) page 3 for a review on the surface energy balance).

**3 Novelty**

**Reviewer2: "It is expected that phase lags would occur between $R_{sd}$ and LE since much energy is stored in the ground surface during the day and then released at night, so it is unclear what the novel aspects of the paper really are." "By not considering G, you get phase lags. . . is there something novel to see here?"**

Reply: The reviewer is unclear about the novelty of our findings and states that we find a phase lag (e.g. to the Latent heat flux but also to Potential evapotranspiration") because we use Incoming Solar Radiation and not Available Energy ($R_n - G$) as reference variable. The argument being that the phase lag we observe is mainly caused due to heat storage in the soil as reflected by the soil heat flux.

We disagree on this perspective. First of all the soil heat flux is not sufficient to buffer the diurnal imbalance caused by solar heat of the land surface. Most of the diurnal imbalance is buffered in the lower atmosphere leading to the development of a convective boundary layer (Oke, 1987). To substantiate our argument we repeated to phase lag analysis with Available Energy ($R_n - G$) as reference variable. The results are attached

in Table 1 of this reply. The table is similar to table 4 of the manuscript. For brevity we report the phase lag in minutes to $R_{sd}$ and to $(R_n - G)$. Overall, there is only a minor difference between the two reference variables. This is to be expected since $R_{sd}$ has the largest diurnal variations of the compents of Available Energy. There is only a minor reduction of phase lag (3 min) with respect to the evapotranspiration estimates. This highlights that the soil heat flux is not the main cause of the observed phase lag of the turbulent heat fluxes.

We believe that our analysis is relevant and we show in the manuscript that the diurnal signature of a phase lag to solar radiation provides a mechanistic insight into the diurnal heat exchange processes of the surface with the atmosphere. While the surface energy balance fluxes show rather small phase lags, the temperatures of the surface, the air and the related vapor pressure deficit of the air show very large phase lags. Including these variables as forcing for models (such as Penman-Monteith) may cause that the predicted fluxes yield a phase lag that is larger than what is typically observed. In contrast the surface to air temperature gradient used in well-established remote sensing based approaches (e.g. Timmermans et al. (2007)) corresponds well in its diurnal phase shift with the observed sensible heat flux and therefore yields a better agreement of the phase lag with observations (see Fig. 7). We did not find a similar analysis and interpretation in the literature, but we are open for suggestions to include further relevant literature during the revision.

We hope that our arguments help to avoid potential misunderstandings which have arisen by the critical comments of the reviewer. We will improve the clarity of the manuscript during revision. The other more minor comments of the reviewer will be addressed in another author reply.

[Figure]

**Table 1.** Calculation of the phase lag of different variables to either incoming solar radiation ($R_{sd}$) or to Available Energy (AE). Nummers in parantheses show the standard deviation of the phase lag.

| Variable | wetdry | PhaseLag(min) to Rsd | PhaseLag(min) to AE |
|---|---|---|---|
| Net Radiation | wet | 1 (3) | -2 (2) |
| Net Radiation | dry | -1 (2) | 0 (1) |
| Soil Heat Flux | wet | -6 (8) | -8 (9) |
| Soil Heat Flux | dry | -0 (8) | 2 (7) |
| Available Energy | wet | 3 (3) | NA (NA) |
| Available Energy | dry | -1 (2) | NA (NA) |
| Sensible Heat Flux | wet | -22 (6) | -25 (7) |
| Sensible Heat Flux | dry | -3 (8) | -3 (8) |
| Incoming Longwave | wet | 133 (84) | 124 (77) |
| Incoming Longwave | dry | 176 (51) | 158 (49) |
| LE BRC | wet | 15 (4) | 11 (3) |
| LE BRC | dry | 3 (12) | 3 (11) |
| LE uncor | wet | 14 (5) | 10 (4) |
| LE uncor | dry | 2 (16) | 3 (14) |
| Priestley-Taylor | wet | 9 (5) | 5 (2) |
| Priestley-Taylor | dry | 6 (4) | 6 (3) |
| Penman-Monteith const. gs | wet | 30 (9) | 25 (6) |
| Penman-Monteith const. gs | dry | 35 (11) | 32 (10) |
| FAO Penman-Monteith | wet | 31 (11) | 26 (9) |
| FAO Penman-Monteith | dry | 31 (12) | 29 (12) |
| LE OSEB | wet | 9 (6) | 5 (4) |
| LE OSEB | dry | -2 (5) | -1 (5) |
| LE TSEB | wet | 9 (5) | 5 (2) |
| LE TSEB | dry | 1 (6) | 1 (4) |
| LE STIC | wet | 20 (19) | 15 (19) |
| LE STIC | dry | 14 (14) | 13 (12) |
| Air Temperature | wet | 130 (41) | 122 (41) |
| Air Temperature | dry | 138 (35) | 130 (37) |
| Surface Temperature | wet | 51 (18) | 46 (16) |
| Surface Temperature | dry | 51 (13) | 49 (13) |
| Ts - Ta | wet | -22 (8) | -24 (10) |
| Ts - Ta | dry | -10 (7) | -7 (7) |
| Vapor Pressure | wet | 125 (188) | 113 (185) |
| Vapor Pressure | dry | 52 (247) | 71 (251) |
| Vapor Pressure Deficit | wet | 145 (39) | 134 (40) |
| Vapor Pressure Deficit | dry | 153 (46) | 144 (47) |

**References**

Ohmura, A.: The development and the present status of energy balance climatology, Journal of the Meteorological Society of Japan. Ser. II, 92, 245–285, 2014.

Oke, T.: Boundary layer climates, Routledge, Londan and New York, 1987.

Timmermans, W. J., Kustas, W. P., Anderson, M. C., and French, A. N.: An intercomparison of the Surface Energy Balance Algorithm for Land (SEBAL) and the Two-Source Energy Balance (TSEB) modeling schemes, Remote Sensing of Environment, 108, 369–384, doi: 10.1016/j.rse.2006.11.028, 2007.

---

## Author Comment (AC2) · 24 Sep 2018

We thank reviewer 1 for his careful review. In our reply we will respond to all comments step by step. We repeat each reviewer comment in bold font, followed by our reply. Changes in the text of the main manuscript are highlighted in blue color.

**Reviewer 1: "The study attempts to assess the model biases in the diurnal cycles of evapotranspiration and to analyze the influence of observed input variables under dry and wet conditions. Much effort has been undertaken to analyze a wealth of observed and modeled data. The approach applied in this study is relatively logical. The findings in the paper may be rational. Therefore, I appreciate**

[Figure]

**the authors' effort to handle such much work. However, I have one concern on the presentation. The paper would be publishable in HESS after minor revisions if the author satisfactorily address my concerns."**

Reply: We thank the reviewer for his assessment.

**Reviewer 1: "Lines 22-30 in page 1, This part should be simplified and keep concise for good read- ability."**

Reply: The reviewer refers to the 2nd paragraph of the abstract which includes most findings of the study in a very condensed way. To improve the readability we simplified and focussed on our main findings and rewrote the paragraph as follows.

*We found remarkable, almost linear relationships of the turbulent heat fluxes with Rsd, which, however, exhibit significant phase lags during wet periods. The diurnal signature of a phase lag to solar radiation provides a mechanistic insight into the diurnal heat exchange processes of the surface with the atmosphere. While the surface energy balance fluxes show rather small phase lags, the temperatures of the surface, the air and the related vapor pressure deficit of the air show very large phase lags. Including these variables as forcing for models (such as vapor pressure deficit in Penman-Monteith based formulations) may cause that the predicted fluxes yield a phase lag that is larger than what is observed. In contrast the surface to air temperature gradient used in well-established remote sensing based approaches corresponds well in its diurnal phase shift with the observed sensible heat flux and therefore yields a better agreement of the phase lag of $\lambda E$ with observations under both, wet and dry conditions. We conclude that the phase lag of surface variables to solar radiation represents a simple, but valuable metric to evaluate and improve the representation of land-atmosphere coupling in land-surface schemes.*

**Reviewer 1: "Lines 20-23 in page 2, This study focus on revealing the model**

**biases of evapotranspi- ration by multivariate metrics, therefore recent literatures should be summarized such as Zhou et al., (2018, published in ACP, doi: 10.5194/acp-18-8113-2018) and Zhou et al., (2017, published in JC, doi: 10.1175/JCLI-D-16-0903.1) that investigated the model biases of regional warming in current reanalysis products and attributed those to the modeled land-atmosphere energy budgets and precipitation frequency."**

Reply: We thank the reviewer for his suggestions on recent literature. The mentioned papers are well suited as references since these use statistical relationships of different model variables, such as temperature and incoming solar radiation (Zhou et al., 2018, 2017). Differences in these relationships between models and observations highlight different sensitivities and helps to evaluate models in a systematic way. Therefore we will include these references in the introduction.

**Reviewer 1: "Section Introduction in pages 2-4, some recent relevant literatures should be sum- marized in the paper, such as van Heerwaarden et al., (2010, published in JC, doi: 10.1175/2010JHM1272.1)"**

Reply: We thank the reviewer for pointing us to the paper by van Heerwaarden et al. (2010). We will update the introduction and include this valueable reference paper.

**Reviewer 1: "Lines 11-21 in page 5, There are other approaches to regress this type of the response. Some reasons of the selection of the Camuffo-Bernardi equation should be provided for good readability."**

Reply: We agree with the reviewer that the choice for the Camuffo-Bernardi model should be better motivated, since we have good reasons to use it. We added the following paragraph to the introduction:

*Here, we choose the Camuffo and Bernardi (1982) model because it provides an objective mea-*

*sure of the magnitude of hysteresis loops and it allows for an assessment of statistical significance. We extend the Camuffo and Bernardi (1982) model in two ways. First, we use incoming solar radiation (Rsd) as reference variable instead of net radiation to estimate the phase lag of surface heat flux observations and models. And secondly, we use a harmonic transformation of the Camuffo and Bernardi (1982) regression model to compare the phase lag of variables with different magnitudes and units. This extension allows to compare the diurnal phase lag signatures of the different model inputs and how these influence the resulting diurnal course of the latent heat flux estimate.*

**Reviewer 1: "Lines 25-end in page 8, The average gap is up to 67 Wm-2 and then the diurnal cycle may has a larger gap. How to quantify the influence of energy balance closure gap (before and after correction) on the magnitude and phase lag in the paper?"**

Reply: To assess the potential impact of the closure method we also computed the phase lag statistics for the non-corrected latent heat flux (see Figure 7, Tables 3 and 4). Results show that the phase lag estimates are very similar showing that the correction does not influence magnitude of the observed phase lags.

To improve the communication of this result we adapted P15L11 in the manuscript as follows:
*The uncorrected observations showed a slightly lower wet-dry difference , highlighting that the method to close the energy balance closure gap does not significantly influence the estimated phase lag.*

**Reviewer 1: "Line 21 in page 22, how to justify the sentence ('These interactions are also affected by soil water availability, as reflected in the phase lags.') in the paper? whether adding related literatures or not?"**

Reply: We replace this sentence with: *We also found that the phase lag of the turbulent heat fluxes is affected by soil water availability.*

**Reviewer 1: "Section Conclusions in pages 25-26, The author should rewrite this part to make its logic smooth. If necessary, some discussion should be added to help the readers understand the importance and advantages of this study."**

Reply: in order to improve the readility of the conclusions we improved the summary of our research setup in the beginning of the conclusions.

*We analyzed the relationship of surface heat fluxes and states to incoming solar radiation at the sub-daily timescale for a temperate grass site which experienced a summer drought. Most variables show significant hysteresis loops which we objectively quantified by a linear component and a non-linear phase lag component using multiple linear regression and harmonic analysis. We then compared these diurnal signatures obtained from observations of an Eddy-Covariance site with commonly used but structurally different approaches to model actual and potential evapotranspiration. The models have been forced by the observational data such that the differences to observations can be attributed to model formulation and signals contained in the input data. In terms of actual evapotranspiration, our results ...*

**References**

van Heerwaarden, C. C., Vilà-Guerau de Arellano, J., Gounou, A., Guichard, F., and Couvreux, F.: Understanding the Daily Cycle of Evapotranspiration: A Method to Quantify the Influence of Forcings and Feedbacks, Journal of Hydrometeorology, 11, 1405–1422, doi: 10.1175/2010JHM1272.1, 2010.

Zhou, C., Wang, K., and Ma, Q.: Evaluation of Eight Current Reanalyses in Simulating Land Surface Temperature from 1979 to 2003 in China, Journal of Climate, 30, 7379–7398, doi: 10.1175/JCLI-D-16-0903.1, 2017.

Zhou, C., He, Y., and Wang, K.: On the suitability of current atmospheric reanalyses for regional

warming studies over China, Atmospheric Chemistry and Physics, 18, 8113–8136, doi:https://doi.org/10.5194/acp-18-8113-2018, 2018.

---

## Author Comment (AC3) · 24 Sep 2018

The second reviewer raised three major aspects (model input, shortwave vs. net radiation, novelty). We already replied to these aspects in a separate author reply (https://doi.org/10.5194/hess-2018-310-AC1) in order to follow up within the open interactive discussion. Unfortunately, the reviewer did not reply to that during the interactive discussion. Our key arguments are presented in the mentioned author reply. In this author reply we will respond to all comments step by step and report how we intend to improve our manuscript to take the comments of the reviewer into account.

We repeat each reviewer comment in bold font, followed by our reply. Changes in the text of the main manuscript are highlighted in blue color.

[Figure]

**Reviewer 2: "The author's objectives of the study were to use measurements of hourly incoming shortwave radiation as an independent forcing of the land-atmosphere exchange and assess the response and phase lags of surface heat fluxes. The authors argue that models of ET should be able to capture the magnitude of hysteretic loops under differ- ent conditions."**

Reply: We agree with this summary of the manuscript.

**Reviewer 2: "The writing is good, and the article is well structured. The major concern I have is that incoming solar radiation (Rsd) is used rather than the available energy (Rn-G). It is expected that phase lags would occur between Rsd and LE since much energy is stored in the ground surface during the day and then released at night, so it is unclear what the novel aspects of the paper really are. All the results are fairly straightforward, but again, they are to be expected based on the study design of using Rsd instead of Rn-G.**
**Additionally, descriptions of what was assumed or used as input to the models, (specifically the PT and FAO-56 PM equations) is not adequate, only Rn is in the PT equation listed, not Rn-G as stated in the original equation, so there could be an error in the analysis. It is unclear if measured G was used in the FAO-56 PM equation, or if it was estimated, and same goes for Rn. Based on the lack of clarity as to what Rn and G model was used in the FAO-56 approach, those results cannot be assessed as is. While there is some good discussion on process, the novelty of the study is lacking."**

Reply: We believe that there are some misunderstandings, which we tried to resolve in our first reply to Reviewer 2, please see (https://doi.org/10.5194/hess-2018-310-AC1). Further author remarks can be found below.

**Reviewer 2: "Perhaps a more useful and/or complementary analysis would be to**

[Figure]

**focus on the hourly distribution of the energy balance closure ratio, and assess the controlling factors of the distribution, if any, as it relates to soil moisture and other conditions."**

Reply: While the analysis of the energy balance closure was not a focus of our work, we actually considered potential impacts by the way the energy balance was closed (instantaneous closures using a daily mean Bowen Ratio). To assess the potential impact of the closure method we also computed the phase lag statistics for the non-corrected latent heat flux (see Figure 7, Tables 3 and 4). Results show that the phase lag estimates are very similar showing that the correction does not influence magnitude of the observed phase lags.

To improve the communication of this result we adapted P15L11 in the manuscript as follows:
*The uncorrected observations showed a slightly lower wet-dry difference , highlighting that the method to close the energy balance closure gap does not significantly influence the estimated phase lag.*

**Reviewer 2: "Pg 2 Line 29-30: LE is strongly correlated with Rsd, not the other way."**

Reply: The text will be adapted.

**Reviewer 2: "Pg 2 Line 21-22: Would be good to give a quick summary of these metrics, and why some are more useful than others if they are to be used or referenced later. This would be good so that when the alternative metric is proposed below the reader has some context."**

Reply: We agree with the reviewer that the introduction needs a better motivation on the existing metrics. Therefore we provide a summary of the different metrics in use

and explain why we are using the metric of a phase lag.

*There is a strong need to investigate and to derive metrics based on comprehensive observation that characterize the whole land surface-atmosphere system (Wulfmeyer et al. 2018). Several authors proposed different multivariate metrics to better evaluate land-atmosphere (L-A) interactions in observations and models. Generally, these metrics explore internal relationships between state variables to better characterize key processes and to guide a more systematic exploration and understanding of model deficiencies. A number of* **metrics** *focus on the diurnal evolution of the* **heat and moisture budgets in the planetary boundary layer** *(e.g., Betts 1992, Santanello et al. 2009, Santanello et al., 2017). Also* **statistical metrics** *exploring the strength of linear relationships between surface heat fluxes and states to surface radiation components have been employed to evaluate the performance of reanalysis with observations (Zhou and Wang 2016, Zhou et al., 2017, 2018). Furthermore, there are* **pattern-based metrics** *which focus on non-linear interactions at the diurnal time scale. Wilson et al., (2003) proposed the method of a diurnal centroid to measure the timing of the surface heat fluxes and their timing difference, which was more recently used by Nelson et al., 2018 to quantify the timing of evapotranspiration under different dryness condition for the FLUXNET dataset. In contrast Matheny et al., 2014 and Zhang et al., 2014 explored the diurnal relationship of the latent heat flux to vapor pressure deficit showing a pronounced hysteresis loop. Zheng et al., 2014 also included air temperature and net radiation as references variables and showed that the hysteresis loops of $\lambda E$ to $D_a$ or $T_a$ are large, while there are only small hysteresis effects when Rn was used. Hysteresis loops have also been found when heat fluxes plotted against net radiation (Camuffo and Bernadi 1982; Mallick et al., 2015), with many studies showing hysteretic loops of the soil heat flux against net radiation (Fuchs and Hadas, 1972; Santanello and Friedl, 2003; Sun et al., 2013). The presence of an hysteresis loop indicates that there is a time dependent non-linear control on the variable of interest, typically induced by heat storage processes. Camuffo and Bernardi (1982) showed that the magnitude and direction of such hysteretic loops can be estimated by a multi-linear regression of the variable of interest against the forcing variables and its first order time-derivative. This simple model allows to estimate storage effects on diurnal (Sun et al. 2013) to seasonal time scales (Duan and Bastiaansen 2017).*

**Reviewer 2: "Pg 3 Line 2: I don't think that the other controls (other than Rn and Rsd) on LE remains unclear. . . it is pretty simple from an energy balance perspective (which is what is being discussed so far in terms of Rn and Rs) . . . LE = Rn – H – G . . . Lots to dig into with H obviously. . . and G, and perhaps that is where some of the controls need more study?"**

Reply: We believe that writing the energy balance with $R_n = \lambda E + H + G$ is sufficient when direct measurements are used. However, when modeling the problem it is clear that all terms may depend on each other. For a mechanistic understanding a full treatmeat of the surface energy balance with explicit treatment of all radiation components is required. Reviewer 1 pointed to a recent study by van Heerwaarden et al. (2010) which discusses the complex interactions of at the surface, the surface layer and the planetary boundary layer, all feeding back on LE. The importance of controls on LE must be considered unclear, since there exist different schemes with different input variables to model LE. Many of the input variable are themselves strongly affected by the land-atmosphere exchange and its feedbacks.

**Reviewer 2: "Page 3 Line 16: Why is Rn not used? Better yet, why isn't Rn-G (available energy) used? I don't see that why Rn (and Rn-G) is not used if the authors are indeed trying to better understand controls on LE. . . longwave radiation is a big component of Rn, and G lags no doubt control some of this hysteresis. I feel that the authors are missing too much energy if they just focus on Rsd.**
**Page 3 Line 23: The PT equation requires Rn, not Rsd, so how can you say you focus on using Rsd, but use the PT equation and not use Rn? Same with the PM equation. Please explain how and if Rs is used in these equations here and you can go into greater detail in the methods if needed. "**

Reply: Our analysis focusses on the diurnal relation of evapotranspiration and relevant

surface energy balance fluxes and states to incoming solar radiation. Since Rsd is independent of the surface, it is an ideal reference to calculate phase lags. We acknowledge that all models which we compare in this study actually use net radiation (Rn) as an input variable, which would also justify to directly use Rn as a reference for the phase lag analysis which is suggested by the reviewer. The differences in the obtained phase lag using Rsd or Rn are not substantial. However, there are sound reasons to rather use Rsd than Rn as reference variable:

- $R_n$ is not independent of the surface conditions

$\rightarrow$ fully coupled models would need to compute $R_n$ by solving the surface energy balance including the turbulent heat fluxes

- It is more consistent to use $R_{sd}$ for the phase lag analysis of other observed surface fluxes and states which are used as input to the models

- for example the phase lag analysis of the vertical temperature gradient (Ts-Ta) would not be useful when Rn is used as reference since the temperature gradient reflects a large part of the net longwave exchange which is part of $R_n$

To better communicate our reasoning we will provide a paragraph in the introduction on the surface energy balance and explain why we are using $R_{sd}$.

**Reviewer 2: "Page 4 Line 7: but RH and VPD is coming from gridded weather data, no? So this is a forcing and outside the evaporation model, correct?"**

Reply: The reviewer mentioned the MOD16 algorithm which was compared with other approaches with surface observations in Yang et al. (2015). That approach uses VPD as an input variable (forcing) which depends on air temperature. While there is some uncertainty when RH and VPD is obtained from coarse reanalysis products instead of

in-situ observations, the main physical argument is that VPD (temperature) of the air cannot resolve the spatial variability of surface water limitation as compared to surface temperature.

**Reviewer 2: "Page 6 Line 28-34: This is concerning since the heat storage in the soil slab above the G plate was estimated rather than measured. Sounds like the estimate didn't con- sider changes in soil moisture, which is a big factor in the potential to store heat within soils. Any errors in the estimate, or bad heat storage measurements could cause "perceived hysteresis" when comparing to other energy balance components. When was the harmonic calibrated, to dry or wet conditions, or both? Did the harmonic behave differently (have different parameters) when assessed during wet vs dry conditions as anticipated?"**

Reply: The total ground heat flux can be obtained by measuring the soil heat flux at a given depth and an correction based on an estimate of heat storage changes above the heat flux plate (Massman 1992). The preferred method for the heat storage changes above the heat flux plate are soil temperature measurements. However, the upper soil temperature sensor failed after two weeks and the following period was characterized by a longer dry period. To circumvent this problem we used an alternative method based on a harmonic transformation of the heat flux plate measurements. The critique of the reviewer is that we did not take the soil moisture dependency of this method into account. This method requires an estimate of the damping depth $D$ which was obtained by the exponential decay of the temperature amplitude of soil temperature measurements.

$D$ is proportional to the square root of the thermal diffusivity and is only weakly dependent on soil moisture for clayey soils above 0.1 m3 m-3 water content (Jury and Horton, 2004) we had at our site. For the present work, $D$ was determined by the exponential decay with depth of the soil temperature amplitude measured for the diurnal cycle in

2, 5, 15, and 30 cm depth at 15 different days between 12th June and 4th July. The mean of $12.27 \pm 0.91$ cm of these determinations was used for harmonic analysis. As mentioned in the manuscript, the upper soil sensors began to fail after 30th June and no determinations of D were possible after 4th July. Ten (five) determinations were performed for soil moisture contents >15% (<15%) where $D$ was obtained to $12.55 \pm 0.65$ cm ($11.71 \pm 1.15$ cm). The differences between the calculated ground heat fluxes using $D = 12.27cm$ and $D = 12.55$ and $D = 11.71cm$, respectively, were always $< 10Wm^{-2}$ so that the used value of $D = 12.27cm$ is a good compromise. For the data until 30th June we find a linear relationship with a slope of 1.05 and $R^2 = 0.94$ for the ground heat flux calculated with harmonic analysis of the HFP fluxes and the heat flux plate method with correction for heat storage. Please also find a figure attached to this reply which shows the diurnal cycles of the total soil heat flux estimates obtained by the upper soil temperature measurements (magenta) and the soil heat flux from the harmonic correction of the soil heat flux plate (blue). The plot only shows sunny days used in the analysis and also reports the top soil moisture of that day. The plots shows higher soil heat fluxes under the wetter conditions for both methods. We thus consider that the total soil heat flux obtained by the harmonic correction of the soil heat flux plate characterizes the diurnal dynamics of the soil heat conduction rather well.

We will add a summary of this explanation to the description in section 2.2.

**Reviewer 2: "Page 8 Line 31: What time step was Qgap (the energy balance closure) assessed? Every 30min?"**

Reply: Yes, the gap has been determined for each time step.

**Reviewer 2: "Page 10 Line 25: The PT equations uses Rn-G, not simply Rn as writ- ten. What did you use? Rn or Rn-G (see equation 14 of the PT paper - ftp://ftp.library.noaa.gov/docs.lib/htdocs/rescue/mwr/100/mwr-100-02-0081.pdf ).**

**The phase lag results can't be interpreted until this is cleared up."**

Reply: We corrected equation 7 to use (Rn-G). This was also used in the analysis, so the interpretation will not change.

**Reviewer 2: "Page 11 Line 14-16: Is Rsd used and then Rn and G is estimated following the procedures of FAO-56, or is the measured Rn-G used? This needs to be spelled out to understand the results."**

Reply: Our strategy is to use all model forcing directly from the observations. The procedures of FAO are only recommended when input data is missing. We added one sentence to make this clear: *All other input variables to equation (8) where directly obtained from the observations.*

**Reviewer 2: "Figure 6. Be consistent calling incoming shortwave Rsd vs Global radiation. . . you say both."**

Reply: We updated the figures labels of Figures 6, 8, 9 and 11 accordingly.

**Reviewer 2: "Figure 7 isn't very useful since it is Rsd on the x, and not Rn-G. I guess I don't see the point since phase lag is to be expected (and greater for wet conditions as shown), and it is unclear how G was considered in the FAO approach."**

Reply: This comment regards the question of using Rsd or Rn as reference to quantify the phase lag. We already replied to this in a separate author reply. With respect to Figure 7, which shows the phase lag of the different latent heat flux estimates against Rsd we find a general wet-dry difference in the observations and most models but not for the Penman-Monteith based approaches. We also computed the phase lag of the

key model input parameters in Fig 12. The reviewer suggested to used Rn-G as a reference. Doing this will not change Figure 7 much and thus also not the conclusions. See also Table 1 of our previous reply https://doi.org/10.5194/hess-2018-310-AC1.

**Reviewer 2: "Page 17 line 4-5: The authors state that "Generally, there was only a small hysteresis in the available energy (Rn – G ) (Table 4)" which is exactly what one would expect if Rn-G was used. So by not including longwave and G there is phase lag, which is to be expected, so I don't see the point of the paper really. . . Also, there would be more phase lag in wet soil conditions, than in dry conditions since heat storage is greater when there is more water in the soil. By not considering G, you get phase lags. . . is there something novel to see here?"**

Reply: We replied to this point in a separate reply. The soil heat flux shows a small phase lag to Rsd which increases in magnitude when wet. However, the phase differences of the turbulent heat fluxes are even larger in magnitude than the ones of G, see Table 4 and Fig.12. These phase lags are also present when Rn-G is used as reference (instead of Rsd). This is consistent with the argument that the soil heat flux is too small to compensate the diurnal imbalance caused by solar radiation. Hence the land-atmosphere heat exchange strongly contributes to balance the large diurnal forcing of solar radiation.

We will put more emphasis on this important point in the revised discussion of the manuscript.

**Reviewer 2: "Page 20 and Figure 11: The results of the hysteresis in humidity variables are what you would expect. The VPD is lowest in the morning, and highest in the mid to late afternoon, and largely a function of es, since it is a fairly humid environment, so what is novel here?"**

Reply: We believe that the diurnal course of VPD is known to most researchers. The key point is that VPD is used as the driving gradient in the Penman-Monteith approaches. This gradient shows a strong hysteresis loop, while the surface to air temperature difference, which is the driving gradient of the energy balance residual approaches shows only a small hysteresis. Visualizing this difference in the two driving gradients (cf. Fig. 9 and Fig 11) should highlight the key differences in these approaches.

**Reviewer 2: "Page 25 Line 22: Yes this was quantified, but it was expected, and it changes in time and space, based on the land surface conditions, and met. forcings."**

Reply: We will update the conclusions of the manuscript to improve the clarity of our writing, see also our reply to reviewer 1 https://doi.org/10.5194/hess-2018-310-AC2.

**Reviewer 2: "Page 25 Line 23: Explain exactly how these results have practical application for re- mote sensing based models? This was never fully described, that is why this phase lag issue is so important for remote sensing studies of LE to consider or include."**

Reply: There are three points which are relevant for remote sensing based approaches:

- We did the comparison for three different remote sensing approaches, highlighting why energy balance approaches are better suited given better agreement in terms of the phase lag. Hence we guide model selection for remote sensing based evapotranspiration retrievals. This is the main contribution of this manuscript.

- The phase lag analysis allows to estimate the magnitude of heat storage changes. This is relevant because heat storage changes in the surface must
also be modeled by remote sensing based approaches.

- The phase lag information can be used to improve sub-daily and daily heat flux estimates from instantaneous observations usually provided from polar-orbiting satellites. Thereby one can use Eq. 1 of the manuscript with knowledge of the phase shift and incoming solar radiation to model the diurnal cycle of the heat flux. This would extend the usual assumption of a constant evaporative fraction over a day (Crago and Brutsaert, 1996; Alfieri et al., 2017).

We will update the discussion of the manuscript to make our contribution for remote sensing based approaches clear.

**Reviewer 2: "Page 25 Line 30-33: There is too little information on the specifics in the paper of FAO-PM approach applied to assess if this is a correct conclusion."**

Reply: As already comment above, we will add information on input variables in the text (in addition to Table 2 summarizing the input data).

**References**

Alfieri, J. G., Anderson, M. C., Kustas, W. P., and Cammalleri, C.: Effect of the revisit interval and temporal upscaling methods on the accuracy of remotely sensed evapotranspiration estimates, Hydrol. Earth Syst. Sci., 21, 83–98, doi:10.5194/hess-21-83-2017, 2017.

Crago, R. and Brutsaert, W.: Daytime evaporation and the self-preservation of the evaporative fraction and the Bowen ratio, Journal of Hydrology, 178, 241–255, doi:10.1016/0022-1694(95)02803-X, 1996.

Jury, W. A. and Horton, R.: Soil Physics, John Wiley & Sons, Inc, 6th edn., 2004.

van Heerwaarden, C. C., Vilà-Guerau de Arellano, J., Gounou, A., Guichard, F., and Couvreux, F.: Understanding the Daily Cycle of Evapotranspiration: A Method to Quantify the

Influence of Forcings and Feedbacks, Journal of Hydrometeorology, 11, 1405–1422, doi:
10.1175/2010JHM1272.1, 2010.

Yang, Y., Long, D., Guan, H., Liang, W., Simmons, C., and Batelaan, O.: Comparison
of three dual-source remote sensing evapotranspiration models during the MUSOEXE-12
campaign: Revisit of model physics, Water Resources Research, 51, 3145–3165, doi:
10.1002/2014WR015619, 2015.

[Figure]

[Figure]

Fig. 1. Comparison of two different estimates of the total soil heat flux for sunny days. Note, that the upper soil temperature sensor was unreliable after 2015-06-30.

[Figure]

---

## Author Response (AR1)

**Point-by-point reply to comments of both reviewers for manuscript submitted to HESS, *hess-2018-310**

By Maik Renner and co-authors

November 21, 2018

This document collects all comments and replies to the reviewers into a single document and highlights how the draft was adapted during revision. Detailed replies to the reviewer comments can be found on the discussion page of the manuscript `https://www.hydrol-earth-syst-sci-discuss.net/hess-2018-310/`.

We repeat each reviewer comment in bold font, followed by our reply. Changes in the text of the main manuscript are highlighted in blue color. A track changes version is attached to this reply, with changes highlighted in dark-orange.

**Title**

We changed the title to be more descriptive of the work and put emphazis into the phase lag analysis.

Old Title: "Understanding model biases in the diurnal cycle of evapotranspiration: a case study in Luxembourg"

New Title: "Using phase lags to evaluate model biases in simulating the diurnal cycle of evapotranspiration: a case study in Luxembourg".

**Detailed replies**

**Reviewer 1: "The study attempts to assess the model biases in the diurnal cycles of evapotranspira- tion and to analyze the influence of observed input variables under dry and wet condi- tions. Much effort has been undertaken to analyze a wealth of observed and modeled data. The approach applied in this study is relatively logical. The findings in the paper may be rational. Therefore, I appreciate the authors' effort to handle such much work. However, I have one concern on the presentation. The paper would be publishable in HESS after minor revisions if the author satisfactorily address my concerns."**

Reply: We thank the reviewer for his careful review. We tried to address all concerns and improve clarity.

**Reviewer 1: "Lines 22-30 in page 1, This part should be simplified and keep concise for good read-ability."**

Reply: The reviewer refers to the abstract which includes most findings of the study in a very condensed way. To improve the readability we simplified and focussed on our main findings and rewrote the abstract as follows:

*Abstract: While modeling approaches of evapotranspiration ($\lambda E$) perform reasonably well when evaluated at daily or monthly time scales, they can show systematic deviations at the sub-daily time scale, which results in potential biases in modeled $\lambda E$ to global climate change. Here we decompose the diurnal variation of heat fluxes and meteorological variables into their direct response to incoming solar radiation (Rsd) and a phase shift to Rsd. We analyze data from an Eddy-Covariance station at a temperate grassland site, which experienced a pronounced summer drought. We employ three structurally different modeling approaches of $\lambda E$, which are used in remote sensing retrievals and quantify how well these models represent the observed diurnal cycle under clear sky conditions. We find that energy balance residual approaches, which use the surface-air temperature gradient as input are able to reproduce the reduction of the phase lag from wet to dry conditions. However, approaches which use the vapor pressure deficit (Da) as driving gradient (Penman-Monteith) show significant deviations from the observed phase lags, which is found to depend on the parameterization of surface conductance to water vapor. This is due to the typically strong phase lag of 2-3h of Da, while the observed phase lag of $\lambda E$ is only in the order of 15 min. In contrast, the temperature gradient shows phase differences in agreement with the sensible heat flux and represents the wet-dry difference rather well. We conclude that phase lags contain important information on the different mechanisms of diurnal heat storage and exchange, and, thus allow a process-based insight to improve the representation of land-atmosphere interactions in models.*

**Reviewer 1:** "Lines 20-23 in page 2, This study focus on revealing the model biases of evapotranspi- ration by multivariate metrics, therefore recent literatures should be summarized such as Zhou et al., (2018, published in ACP, doi: 10.5194/acp-18-8113-2018) and Zhou et al., (2017, published in JC, doi: 10.1175/JCLI-D-16-0903.1) that investigated the model biases of regional warming in current reanalysis products and attributed those to the modeled land-atmosphere energy budgets and precipitation frequency."

Reply: We thank the reviewer for his suggestions on recent literature. The mentioned papers are well suited as references since these use statistical relationships of different model variables, such as temperature and incoming solar radiation (Zhou et al., 2018, 2017). Differences in these relationships between models and observations highlight different sensitivities and helps to evaluate models in a systematic way. We included these references in the introduction:

*Also statistical metrics exploring the strength of linear relationships between surface heat fluxes and states to surface radiation components have been employed to evaluate the performance of reanalysis with observations (Zhou and Wang 2016, Zhou et al., 2017, 2018).*

**Reviewer 1:** "Section Introduction in pages 2-4, some recent relevant literatures should be sum- marized in the paper, such as van Heerwaarden et al., (2010, published in JC, doi: 10.1175/2010JHM1272.1)"

Reply: We thank the reviewer for pointing us to the paper by van Heerwaarden et al. (2010). We updated the introduction and included this valueable reference paper as follows:

*These interactions are particularly dominant at the diurnal time scale (e.g. De Bruin and Holtslag 1982) and depend on meteorological as well as on surface conditions (Jarvis and McNaughton, 1986; van Heerwaarden et al., 2010).*

**Reviewer 1:** "Lines 11-21 in page 5, There are other approaches to regress this type of the response. Some reasons of the selection of the Camuffo-Bernardi equation should be provided for good readability."

Reply: We agree with the reviewer that the choice for the Camuffo-Bernardi model should be better motivated, since we have good reasons to use it. We added the following paragraph to the introduction:

*Here, we choose the Camuffo and Bernardi (1982) model because it provides an objective measure of the magnitude of hysteresis loops and it allows for an assessment of statistical significance. We extend the Camuffo and Bernardi (1982) model in two ways. First, we use incoming solar radiation (Rsd) as reference variable instead of net radiation to estimate the phase lag of surface heat flux observations and models. And secondly, we use a harmonic transformation of the Camuffo and Bernardi (1982) regression model to estimate the phase lag in time units. This extension allows to compare the diurnal phase lag signatures of the different model inputs and how these influence the resulting diurnal course of the latent heat flux estimate.*

**Reviewer 1:** "Lines 25-end in page 8, The average gap is up to 67 Wm-2 and then the diurnal cycle may has a larger gap. How to quantify the influence of energy balance closure gap (before and after correction) on the magnitude and phase lag in the paper?"

Reply: To assess the potential impact of the closure method we also computed the phase lag statistics for the non-corrected latent heat flux (see Figure 7, Tables 3 and 4). Results show that the phase lag estimates are very similar showing that the correction does not influence magnitude of the observed phase lags.

To improve the communication of this result we adapted P15L11 in the manuscript as follows: *The uncorrected observations showed only a slightly lower wet-dry difference, highlighting that the method to close the energy balance closure gap does not significantly influence the estimated phase lag.*

We also identified a typo in reporting the average energy balance closure gap on P8L31. It is $37 \, Wm^{-2}$ instead of $67 \, Wm^{-2}$. We updated the text as follows: *For our site we observed on average a slope of $(H+\lambda E)\,(Rn\breve{} G) = 0.81$ (by linear regression) with an average gap of $37 \, Wm^{-2}$ over the whole duration of the field campaign.*

**Reviewer 1:** "Line 21 in page 22, how to justify the sentence ('These interactions are also affected by soil water availability, as reflected in the phase lags.') in the paper? whether adding related literatures or not?"

Reply: We replace this sentence with: *We also found that the phase lag of the turbulent heat fluxes is affected by soil water availability.*

**Reviewer 1:** "Section Conclusions in pages 25-26, The author should rewrite this part to make its logic smooth. If necessary, some discussion should be added to help the readers understand the importance and advantages of this study."

Reply: We agree with the reviewer and revised the first paragraph of the conclusions.

*We analyzed the relationship of surface heat fluxes and states to incoming solar radiation at the sub-daily timescale for a temperate grassland site which experienced a summer drought. Most variables showed significant hysteresis loops which we objectively quantified by a linear component and a non-linear phase lag component using*

*multiple linear regression and harmonic analysis. We then compared these diurnal signatures obtained from observations of an Eddy-Covariance station with commonly used but structurally different approaches to model actual and potential evapotranspiration. The models have been forced by the observational data such that the differences to observations can be attributed to model formulation and signals contained in the input data. Our analysis guides model selection with a preference for the temperature gradient approaches, because the vertical temperature gradient contains relevant signals of soil moisture limitation as opposed to the vapor pressure deficit of the air.*

**Replies to reviewer 2**

**Reviewer 2: "The author's objectives of the study were to use measurements of hourly incoming shortwave radiation as an independent forcing of the land-atmosphere exchange and assess the response and phase lags of surface heat fluxes. The authors argue that models of ET should be able to capture the magnitude of hysteretic loops under differ- ent conditions."**

Reply: We agree with this summary of the manuscript.

**Reviewer 2: "The writing is good, and the article is well structured. The major concern I have is that incoming solar radiation (Rsd) is used rather than the available energy (Rn-G)."**

Reply: Our main reasoning is that available energy is not an independent variable as it depends on surface temperature. We added a paragraph in the introduction to explain our reasoning:

*We specifically choose incoming solar radiation Rsd as the reference for the phase shift analysis, since Rsd can be regarded as an independent forcing of the surface energy balance (e.g. Ohmura (2014)):*

$$R_{sd}(1 - \alpha) + R_{ld} - H - \lambda E - G = \sigma T^4 + m \tag{1}$$

*With surface albedo $\alpha$, incoming longwave radiation Rld, sensible heat flux H, latent heat flux $\lambda E$, the conductive soil heat flux G, the outgoing longwave radiation $\sigma T^4$ and storage terms of the surface layer summarized in m. This form of the surface energy balance provides the direction of the energy exchange processes at the surface, illustrating that the terms on the right-hand side depend on heat fluxes on the left-hand side of Eq. (1) (Ohmura 2014). As a consequence, the term net radiation Rn, which resembles the radiation budget of the shortwave and longwave components: $R_n = R_{sd}(1 - \alpha) + R_{ld} - \sigma T^4$, cannot be regarded as an independent surface forcing. Therefore, we prefer to use Rsd instead of Rn or Rn-G as the reference variable for phase shift analysis of the latent heat flux and the main input variables of evapotranspiration model approaches.*

**Reviewer 2: ""It is expected that phase lags would occur between $R_{sd}$ and LE since much energy is stored in the ground surface during the day and then released at night, so it is unclear what the novel aspects of the paper really are." "By not considering G, you get phase lags. . . is there something novel to see here?""**

Reply: The reviewer is unclear about the novelty of our findings and states that we find a phase lag (e.g. to the Latent heat flux but also to Potential evapotranspiration") because we use Incoming Solar Radiation and not Available Energy $(R_n - G)$ as reference variable. The argument being that the phase lag we observe is mainly caused due to heat storage in the soil as reflected by the soil heat flux.

We disagree on this perspective. First of all the soil heat flux is not sufficient to buffer the diurnal imbalance caused by solar heat of the land surface. Most of the diurnal imbalance is buffered in the lower atmosphere leading to the development of a convective boundary layer (Oke, 1987). To substantiate our argument we repeated to phase lag analysis with Available Energy $(R_n - G)$ as reference variable. We added the resulting phase lag as another column to Table 4 of the manuscript. Overall, there is only a minor difference of the phase lag between the two reference variables. This is to be expected since $R_{sd}$ has the largest diurnal variations of the components of Available Energy. There is only a minor reduction of phase lag (3 min) with respect to the evapotranspiration estimates. This highlights that the soil heat flux is not the main cause of the observed phase lag of the turbulent heat fluxes.

**Reviewer 2: "Page 3 Line 16: Why is Rn not used? Better yet, why isn't Rn-G (available energy) used? I don't see that why Rn (and Rn-G) is not used if the authors are indeed trying to better understand controls on LE. . . longwave radiation is a big component of Rn, and G lags no doubt control some of this hysteresis. I feel that the authors are missing too much energy if they just focus on Rsd.**
**Page 3 Line 23: The PT equation requires Rn, not Rsd, so how can you say you focus on using Rsd, but use the PT equation and not use Rn? Same with the PM equation. Please explain how and if Rs is used in these equations here and you can go into greater detail in the methods if needed. "**

Reply: Our analysis focusses on the diurnal relation of evapotranspiration and relevant surface energy balance fluxes and states to incoming solar radiation. Since Rsd is independent of the surface, it is an ideal reference to calculate phase lags. We acknowledge that all models which we compare in this study actually use available energy (net radiation - ground heat flux) (Rn-G) as an input variable, which would also justify to directly use Rn as a reference for the phase lag analysis which is suggested by the reviewer. The differences in the obtained phase lag using Rsd or Rn are not substantial, see updated Table 4.

We added a paragraph to the results section 3.3: *Since all evapotranspiration schemes use Rn-G as forcing, we also computed that phase lags with Rn-G as reference variable (see Table 4). The difference to Rsd as reference are, however, rather small with slightly lower phase lags and in the range of the standard deviation of the daily estimates. This is because there is hardly any phase lag between Rsd and Rn and because the magnitude and the phase lag of the soil heat flux is rather small.*

We also think that this result needs to be better discussed and added a paragraph to the first section of the discussion: *It is important to emphasize here, that the phase lags we found here are not dominated by diurnal heat storage changes below the surface. The phase lag of the soil heat flux to Rsd is smaller than the phase shifts of the turbulent heat fluxes. All models we employ here use available energy (Rn – G) as input to estimate LE. So one may think that the identified phase lags are due to choosing Rsd as the reference. However, the phase lag of the latent heat flux would only reduce by about 3 min when one would choose Rn-G instead of Rsd as reference variable to calculate the phase lags. This is because there is almost no phase lag between Rn and Rsd and the fact that both the magnitude and the phase lag of the soil heat flux are relatively small.*

**Reviewer 2: "Additionally, descriptions of what was assumed or used as input to the models, (specifically the PT and FAO-56 PM equations) is not adequate, only Rn is in the PT equation listed, not Rn-G as stated in the original equation, so there could be an error in the analysis. It is unclear if measured G was used in the FAO-56 PM equation, or if it was estimated, and same goes for Rn. Based on the lack of clarity as to what Rn and G model was used in the FAO-56 approach, those results cannot be assessed as is. While there is some good discussion on process, the novelty of the study is lacking."**

Reply: We believe that there are some misunderstandings, which we tried to resolve in our first reply to Reviewer 2, please see (`https://doi.org/10.5194/hess-2018-310-AC1`). All models have been driven by the observational data which is important because this allows a fair comparison between models. A list of input is provided in Table 2 of the manuscript.

Specifically, both potential evapotranspiration estimates, the Priestley-Taylor evapotranspiration and FAO Penman-Monteith estimate, use available energy $(R_n - G)$ as input. We apologize that the soil heat flux was missing in the Priestley-Taylor Equation (Eq. 7) and corrected this typo in the revision. The calculations are, however, not affected by this typo. Also, the FAO Penman-Monteith equation was driven with net radiation and soil heat flux from the observations and not from one of the empirical replacements as provided in the FAO-56. Hence we can assure that the findings of systematically different diurnal cycles of the Penman-Monteith driven models is indeed related to the model formulation and not to errors in the analysis.

All code (and data) to reproduce the analysis are provided in a public accessible repository with this revision of the manuscript.

**Reviewer 2: "Perhaps a more useful and/or complementary analysis would be to focus on the hourly distribution of the energy balance closure ratio, and assess the controlling factors of the distribution, if any, as it relates to soil moisture and other conditions."**

Reply: While the analysis of the energy balance closure was not a focus of our work, we actually considered potential impacts by the way the energy balance was closed (instantaneous closures using a daily mean Bowen Ratio). To assess the potential impact of the closure method we also computed the phase lag statistics for the non-corrected latent heat flux (see Figure 7, Tables 3 and 4). Results show that the phase lag estimates are very similar showing that the correction does not influence magnitude of the observed phase lags.

To improve the communication of this result we adapted P15L11 in the manuscript as follows: *The uncorrected observations showed only a slightly lower wet-dry difference, highlighting that the method to close the energy balance closure gap does not significantly influence the estimated phase lag.*

**Reviewer 2: "Pg 2 Line 29-30: LE is strongly correlated with Rsd, not the other way."**
Reply: The text was adapted.

**Reviewer 2: "Pg 2 Line 21-22: Would be good to give a quick summary of these metrics, and why some are more useful than others if they are to be used or referenced later. This would be good so that when the alternative metric is proposed below the reader has some context."**
Reply: We agree with the reviewer that the introduction needs a better motivation on the existing metrics.

Therefore we provide a summary of the different metrics in use and explain why we are using the metric of a phase lag.

*There is a strong need to investigate and to derive metrics based on comprehensive observation that characterize the whole land surface-atmosphere system (Wulfmeyer et al. 2018). Several authors proposed different multivariate metrics to better evaluate land-atmosphere (L-A) interactions in observations and models. Generally, these metrics explore internal relationships between state variables to better characterize key processes and to guide a more systematic exploration and understanding of model deficiencies. A number of **metrics** focus on the diurnal evolution of the **heat and moisture budgets in the planetary boundary layer** (e.g., Betts 1992, Santanello et al. 2009, Santanello et al., 2017). Also **statistical metrics** exploring the strength of linear relationships between surface heat fluxes and states to surface radiation components have been employed to evaluate the performance of reanalysis with observations (Zhou and Wang 2016, Zhou et al., 2017, 2018). Furthermore, there are **pattern-based metrics** which focus on non-linear interactions at the diurnal time scale. Wilson et al., (2003) proposed the method of a diurnal centroid to measure the timing of the surface heat fluxes and their timing difference, which was more recently used by Nelson et al., 2018 to quantify the timing of evapotranspiration under different dryness condition for the FLUXNET dataset. In contrast Matheny et al., 2014 and Zhang et al., 2014 explored the diurnal relationship of the latent heat flux to vapor pressure deficit showing a pronounced hysteresis loop. Zheng et al., 2014 also included air temperature and net radiation as references variables and showed that the hysteresis loops of $\lambda E$ to $D_a$ or $T_a$ are large, while there are only small hysteresis effects when Rn was used. Hysteresis loops have also been found when heat fluxes plotted against net radiation (Camuffo and Bernadi 1982; Mallick et al., 2015), with many studies showing hysteretic loops of the soil heat flux against net radiation (Fuchs and Hadas, 1972; Santanello and Friedl, 2003; Sun et al., 2013). The presence of an hysteresis loop indicates that there is a time dependent non-linear control on the variable of interest, typically induced by heat storage processes. Camuffo and Bernardi (1982) showed that the magnitude and direction of such hysteretic loops can be estimated by a multi-linear regression of the variable of interest against the forcing variables and its first order time-derivative. This simple model allows to estimate storage effects on diurnal (Sun et al. 2013) to seasonal time scales (Duan and Bastiaansen 2017).*

**Reviewer 2: "Pg 3 Line 2: I don't think that the other controls (other than Rn and Rsd) on LE remains unclear. . . it is pretty simple from an energy balance perspective (which is what is being discussed so far in terms of Rn and Rs) . . . LE = Rn − H − G . . . Lots to dig into with H obviously. . . and G, and perhaps that is where some of the controls need more study?"**

Reply: We believe that writing the energy balance with $R_n = \lambda E + H + G$ is sufficient when direct measurements are used. However, when modeling the problem it is clear that all terms may depend on each other. For a mechanistic understanding a full treatment of the surface energy balance with explicit treatment of all radiation components is required. Reviewer 1 pointed to a recent study by van Heerwaarden et al. (2010) which discusses the complex interactions of at the surface, the surface layer and the planetary boundary layer, all feeding back on LE. The importance of controls on LE must be considered unclear, since there exist different schemes with different input variables to model LE. Many of the input variable are themselves strongly affected by the land-atmosphere exchange and its feedbacks.

**Reviewer 2: "Page 4 Line 7: but RH and VPD is coming from gridded weather data, no? So this is a forcing and outside the evaporation model, correct?"**

Reply: The reviewer mentioned the MOD16 algorithm which was compared with other approaches with surface observations in Yang et al. (2015). That approach uses VPD as an input variable (forcing) which depends on air temperature. While there is some uncertainty when RH and VPD is obtained from coarse reanalysis products instead of in-situ observations, the main physical argument is that VPD (temperature) of the air cannot resolve the spatial variability of surface water limitation as compared to surface temperature.

**Reviewer 2: "Page 6 Line 28-34: This is concerning since the heat storage in the soil slab above the G plate was estimated rather than measured. Sounds like the estimate didn't con- sider changes in soil moisture, which is a big factor in the potential to store heat within soils. Any errors in the estimate, or bad heat storage measurements could cause "perceived hysteresis" when comparing to other energy balance components. When was the harmonic calibrated, to dry or wet conditions, or both? Did the harmonic behave differently (have different parameters) when assessed during wet vs dry conditions as anticipated?"**

Reply: The total ground heat flux can be obtained by measuring the soil heat flux at a given depth and an correction based on an estimate of heat storage changes above the heat flux plate (Massman 1992). The preferred method for the heat storage changes above the heat flux plate are soil temperature measurements. However, the upper soil temperature sensor failed after two weeks and the following period was characterized by a longer dry period. To circumvent this problem we used an alternative method based on a harmonic transformation of the

heat flux plate measurements. The critique of the reviewer is that we did not take the soil moisture dependency of this method into account. This method requires an estimate of the damping depth $D$ which was obtained by the exponential decay of the temperature amplitude of soil temperature measurements.

$D$ is proportional to the square root of the thermal diffusivity and is only weakly dependent on soil moisture for clayey soils above 0.1 m3 m-3 water content (Jury and Horton, 2004) we had at our site. For the present work, $D$ was determined by the exponential decay with depth of the soil temperature amplitude measured for the diurnal cycle in 2, 5, 15, and 30 cm depth at 15 different days between 12th June and 4th July. The mean of $12.27 \pm 0.91$ cm of these determinations was used for harmonic analysis. As mentioned in the manuscript, the upper soil sensors began to fail after 30th June and no determinations of D were possible after 4th July. Ten (five) determinations were performed for soil moisture contents >15% (<15%) where $D$ was obtained to $12.55 \pm 0.65$ cm ($11.71 \pm 1.15$ cm). The differences between the calculated ground heat fluxes using $D = 12.27 cm$ and $D = 12.55$ and $D = 11.71 cm$, respectively, were always $< 10 W m^{-2}$ so that the used value of $D = 12.27 cm$ is a good compromise. For the data until 30th June we find a linear relationship with a slope of 1.05 and $R^2 = 0.94$ for the ground heat flux calculated with harmonic analysis of the HFP fluxes and the heat flux plate method with correction for heat storage. Please also find a figure attached to this reply which shows the diurnal cycles of the total soil heat flux estimates obtained by the upper soil temperature measurements (magenta) and the soil heat flux from the harmonic correction of the soil heat flux plate (blue). The plot only shows sunny days used in the analysis and also reports the top soil moisture of that day. The plots shows higher soil heat fluxes under the wetter conditions for both methods. We thus consider that the total soil heat flux obtained by the harmonic correction of the soil heat flux plate characterizes the diurnal dynamics of the soil heat conduction rather well.

We will added a summary of this explanation to the description in section 2.2: *Unfortunately, the two upper temperature probes and soil matric potential sensors showed data gaps and erroneous values from 30 June until excavation on 23 July, 2015. Thus, the ground heat flux was calculated by the heat flux plate method with correction for heat storage (Massman, 1992) only for the period from 11 June to 30 June, 2015. To still obtain soil heat fluxes for the entire measuring period, additionally harmonic wave analysis (Duchon and Hale, 2012) of the heat flux plate data was applied. The harmonic wave analysis calculates the wave spectrum at the soil surface from the Fourier transform of the soil heat flux measured by the heat flux plates in a few cm depth (here: 8 cm) by correcting for wave amplitude damping and phase shift. The surface ground heat flux is then obtained by an inverse Fourier transformation of the corrected wave spectrum. The method has a dependence on soil moisture affecting the damping depth. The dependence is, however, weak for clayey soils with soil water contents > 10% (Jury and Horton 2004) as observed at the site. The damping depth was obtained by the exponential decay of the soil temperature amplitude measured at the various depths. Differences in the damping depth between wet and drier soil moisture conditions only yielded differences in G smaller than 10 W m-2. Therefore, we used a constant damping depth for the whole period.*

**Reviewer 2: "Page 8 Line 31: What time step was Qgap (the energy balance closure) assessed? Every 30min?"**

Reply: Yes, the gap has been determined for each time step.

*To correct the turbulent fluxes for the energy balance closure gap (evaluated at the 30 min time steps), we use a correction based on the Bowen ratio (BR) (Twine et al., 2000), which is directly related to the evaporative fraction fE = 1/(BR+1) to obtain corrected fluxes*

**Reviewer 2: "Page 10 Line 25: The PT equations uses Rn-G, not simply Rn as writ- ten. What did you use? Rn or Rn-G (see equation 14 of the PT paper - ftp://ftp.library.noaa.gov/docs.lib/htdocs/rescue/mwr/100/mwr-100-02-0081.pdf ). The phase lag results can't be interpreted until this is cleared up."**

Reply: We corrected equation 7 to use (Rn-G). This was also used in the analysis, so the interpretation will not change.

**Reviewer 2: "Page 11 Line 14-16: Is Rsd used and then Rn and G is estimated following the procedures of FAO-56, or is the measured Rn-G used? This needs to be spelled out to understand the results."**

Reply: Our strategy is to use all model forcing directly from the observations. The procedures of FAO are only recommended when input data is missing. We added one sentence to make this clear: *. Here we use the latter definitions of the conductances and use direct measurements for the other input variables to equation (9) to obtain the FAO Penman-Monteith estimate.*

**Reviewer 2: "Figure 6. Be consistent calling incoming shortwave Rsd vs Global radiation. . . you say both."**

Reply: Thank you for pointing this out. We updated the figures labels of Figures 6, 8, 9 and 11 accordingly.

**Reviewer 2:** "Figure 7 isn't very useful since it is Rsd on the x, and not Rn-G. I guess I don't see the point since phase lag is to be expected (and greater for wet conditions as shown), and it is unclear how G was considered in the FAO approach."

Reply: This comment regards the question of using Rsd or Rn as reference to quantify the phase lag. We already replied to this in a separate author reply. With respect to Figure 7, which shows the phase lag of the different latent heat flux estimates against Rsd we find a general wet-dry difference in the observations and most models but not for the Penman-Monteith based approaches. We also computed the phase lag of the key model input parameters in Fig 12. The reviewer suggested to used Rn-G as a reference. Doing this will not change Figure 7 much and thus also not the conclusions. See also the updated Table 4.

**Reviewer 2:** "Page 17 line 4-5: The authors state that "Generally, there was only a small hysteresis in the available energy (Rn − G ) (Table 4)" which is exactly what one would expect if Rn-G was used. So by not including longwave and G there is phase lag, which is to be expected, so I don't see the point of the paper really. . . Also, there would be more phase lag in wet soil conditions, than in dry conditions since heat storage is greater when there is more water in the soil. By not considering G, you get phase lags. . . is there something novel to see here?"

Reply: We already replied to this point. The soil heat flux shows a small phase lag to Rsd which increases in magnitude when wet. However, the phase differences of the turbulent heat fluxes are even larger in magnitude than the ones of G, see Table 4 and Fig.12. These phase lags are also present when Rn-G is used as reference (instead of Rsd). This is consistent with the argument that the soil heat flux is too small to compensate the diurnal imbalance caused by solar radiation. Hence the land-atmosphere heat exchange strongly contributes to balance the large diurnal forcing of solar radiation.

**Reviewer 2:** "Page 20 and Figure 11: The results of the hysteresis in humidity variables are what you would expect. The VPD is lowest in the morning, and highest in the mid to late afternoon, and largely a function of es, since it is a fairly humid environment, so what is novel here?"

Reply: We believe that the diurnal course of VPD is known to most researchers. The key point is that VPD is used as the driving gradient in the Penman-Monteith approaches. This gradient shows a strong hysteresis loop, while the surface to air temperature difference, which is the driving gradient of the energy balance residual approaches shows only a small hysteresis. Visualizing this difference in the two driving gradients (cf. Fig. 9 and Fig 11) should highlight the key differences in these approaches.

**Reviewer 2:** "Page 25 Line 22: Yes this was quantified, but it was expected, and it changes in time and space, based on the land surface conditions, and met. forcings."

Reply: We updated the conclusions of the manuscript to improve the clarity of our writing, see also our reply to reviewer 1 above and `https://doi.org/10.5194/hess-2018-310-AC2`.

**Reviewer 2:** "Page 25 Line 23: Explain exactly how these results have practical application for re- mote sensing based models? This was never fully described, that is why this phase lag issue is so important for remote sensing studies of LE to consider or include."

Reply: The key contribution of the phase lag analysis is that it guides model selection for observational driven LE retrievals, such as remote sensing based approaches. This includes the common measurements for the input of these approaches. As we clearly show they have different phase lags (Fig. 12) and thus influence the resulting LE estimates. We discuss these issues in depth and refer to it in the conclusions. Please note the revised conclusions section, discussed above.

**Reviewer 2:** "Page 25 Line 30-33: There is too little information on the specifics in the paper of FAO-PM approach applied to assess if this is a correct conclusion."

Reply: As already comment above, we added information on input variables in the text (in addition to Table 2 summarizing the input data).

**Other changes**

During revision we updated the coefficients of the empirical Magnus equation to calculate the saturation vapor pressure curve and its temperature derivative, see section 2.2.1. We now use more recent coefficients published by Alduchov and Eskridge (1996). This effects the vapor pressure and saturation vapor pressure estimates and thus the Priestley-Taylor and the Penman-Monteith potential evapotranspiration estimates. The changes are, however, very minor (e.g. change in mean LE PM-FAO in Table 3, approx. $1Wm^{-2}$). Tables 3 and 4 and Figures 4-12 have

been updated.

[revised manuscript text omitted]